# Rate-selected growth of ultrapure semiconducting carbon nanotube arrays

Zhenxing Zhu [1,2], Nan Wei[3], Weijun Cheng[4], Boyuan Shen[1], Silei Sun[1], Jun Gao[1], Qian Wen[1], Rufan Zhang [1], Jun Xu[4], Yao Wang[1] & Fei Wei[1]*

Carbon nanotubes (CNTs) are promising candidates for smart electronic devices. However, it is challenging to mediate their bandgap or chirality from a vapor-liquid-solid growth process. Here, we demonstrate rate-selected semiconducting CNT arrays based on interlocking between the atomic assembly rate and bandgap of CNTs. Rate analysis confirms the Schulz-Flory distribution which leads to various decay rates as length increases in metallic and semiconducting CNTs. Quantitatively, a nearly ten-fold faster decay rate of metallic CNTs leads to a spontaneous purification of the predicted 99.9999% semiconducting CNTs at a length of 154 mm, and the longest CNT can be 650 mm through an optimized reactor. Transistors fabricated on them deliver a high current of 14 $\mu$A $\mu$m$^{-1}$ with on/off ratio around $10^8$ and mobility over 4000 cm$^2$ V$^{-1}$ s$^{-1}$. Our rate-selected strategy offers more freedom to control the CNT purity in-situ and offers a robust methodology to synthesize perfectly assembled nanotubes over a long scale.

---

[1] Beijing Key Laboratory of Green Chemical Reaction Engineering and Technology, Department of Chemical Engineering, Tsinghua University, Beijing 100084, China. [2] Center for Nano and Micro Mechanics, Tsinghua University, Beijing 100084, China. [3] Nano Materials Group, Department of Applied Physics and Center for New Materials, School of Science, Aalto University, PO Box 15100FI-00076 Aalto, Finland. [4] National Laboratory for Information Science and Technology, Institute of Microelectronics, Tsinghua University, Beijing 100084, China. *email: wf-dce@tsinghua.edu.cn

Fundamental studies of charge flow through perfect individual CNTs have revealed prominent characteristics, including superb short channel control enabled by the intrinsic ultrathin body, high output current resulting from high saturation velocity and low resistance contributed by end-bonded contacts[1]. These tubes are considered the most promising materials to replace Si for ultra-scaled logic device applications in the post-Moore's era[2], as demonstrated in the continuous progress on high-performance CNT electronics[3–5]. U.S. military's Defense Advanced Research Projects Agency (DARPA) has recently predicted the forthcoming electronics resurgence and sponsored to fabricate faster and smaller CNT chips[6]. However, such applications may not be accomplished until crucial issues such as precise atom alignment and controlled synthesis based on bandgap and diameter would be resolved. This is because different CNTs possess various chiral structures, characterized by indices $(n, m)$, which enable them with metallic or semiconducting behavior. Although metallic CNTs (m-CNTs) may serve as interconnects with high current-carrying capacities, they lead to high off current and shorted transistors[1]. Adequate strategies for growing aligned semiconducting CNTs (s-CNTs) are highly demanding to accomplish the daunting requirements of purity as high as 99.9999%[7]. Numerous technological tube sorting processes based on bulk-grown CNTs have delivered a large fraction of s-CNTs[3,4]. Whereas, these agglomerated or vertically aligned tubes are short in length with intrinsic structural defects, and subsequent purification, sonication, or separation procedures will cause further damage and overlapping on tubes. Consequently, they will degrade the mobility or trigger the interfered currents in transistors[8]. In contrast, surface-grown ultralong CNTs follow the kite mechanism[9] and their growth is independent of interactions with the substrate and tube-tube effect. The reported longest length extended to 550 mm with more than ten billion carbon dimers C2 being precisely assembled along one chiral direction[10]. Only on these perfect tubes will ballistic transport be easier to achieve due to a lack of defect-site-induced carrier scattering, and devices integrated on these individual long tubes can possess the best uniformity of performances[11]. However, their production is very low, especially for the longer tubes. According to the Schulz-Flory (SF) distribution[10], although it's an equal probability event that if an individual CNT can still be alive after each C2 addition, different batches of aligned CNTs may possess different catalyst activity probabilities $\alpha$ ($\alpha$ describes the probability that the catalysts can keep active enough to support a CNT growth when adding a unit length). The number of ultralong CNTs will exponentially decrease as length increases for whichever $\alpha$. A lower $\alpha$ means a lower percentage of the longer CNTs. Recent progress on these longer CNTs endorsed a 93% selectivity of s-CNTs at a length of 30 mm whereas no obvious semiconducting selectivity was discovered for the infant tubes[12,13]. Impeded by the difficulties in obtaining even longer CNTs, there is lack of evidence how the selectivity of s-CNTs would change with length. It also remains ambiguous how these ultralong CNTs can keep perfectly atomic assembly over a wide length scale. It's important to address these issues in fabricating high-performance devices with these ultralong CNTs and developing industrial applications in carbon-based electronics.

In this study, we demonstrate that m- and s-CNTs respectively follow the SF distribution with different decay rates of their quantities in a natural growth process. Interlocking with their bandgap, the atomic assembly rate of s-CNTs is ten times higher than that of m-CNTs, which leads to a rate-selected growth of the predicted 99.9999% perfect s-CNTs when the length is over 154 mm. Our method provides a strategy to in situ control the s-CNT purity, and also reveals a template mechanism to achieve structural control for CNTs of the vapor-liquid-solid growth mode.

## Results

**Different decay rates of aligned m- and s-CNTs.** We performed careful micro-Raman measurements on free-standing ultralong CNTs, statistically counting the m- and s-tubes at a different length. Tangential vibration (G mode) profiles (Supplementary Fig. 1) demonstrated decreasing Breit-Wigner-Fano (BWF) lineshapes featuring m-CNTs as length increased, tandem with diminished D bands (~1350 cm$^{-1}$) characteristic of defects. Both modes vanished at an approximate length of 50 mm, indicating the decay of the quantities of metallic and defective CNTs. Further statistics on four samples (Supplementary Fig. 2) from different batches all addressed respective decay rates of m- and s-CNTs. We describe the decay rate with half-length $L_{0.5}$, which denotes the length where the CNT quantity ($N_L$) decreases by half compared to that ($N_0$) near the catalyst. Summing up the counts of m- or s-CNTs at the same length from four samples (Fig. 1a) showed quantitatively the $L_{0.5}$ of s-CNTs (74.4 mm) was ~10 times longer than m-CNTs (7.16 mm), acknowledging an order of magnitude higher decay rate of aligned m-CNTs.

One possible reason of the decayed m-CNTs involves a transformation to s-tubes. Nevertheless, massive long s-tubes, indicated by Lorentz TO modes (Fig. 1d), each displayed a uniform color under the illumination of supercontinuum laser (see Fig. 1b and Supplementary Fig. 3)[14]. Another sign showed close radial breathing mode (RBM) frequencies at the top and bottom of each tube (Fig. 1c). Both pieces of evidence validate consistent chiral structures. Thus, it is nontrivial for the case of m- to s-tube transformation. The other occasion describes an intrinsically slow growth rate of m-CNTs and average lengths of the longest ten m- or s-tubes from the same batch were measured at the varied time for validation. As shown in Fig. 1e, stable growth rates of m- and s-CNT populations exemplified two independent kinetic processes. The growth rate of s-CNTs (~80 μm s$^{-1}$) was demonstrated ~10 times faster than m-CNTs (~7 μm s$^{-1}$), which coincided with the decay rate ratio of m-/s-aligned tubes. Therefore, other than the total CNTs, both m- and s-CNTs also follow the SF distribution characteristic of their kinetics-controlled polymerization and stable growth rates. The differences among these three SF distributions lie in the $L_{0.5}$ difference as shown by the Eq. (1)

$$\ln N_L = \ln N_0 - \frac{\ln 2}{L_{0.5} - 1}(L - 1) \tag{1}$$

The larger $L_{0.5,s}$ will lead to a slower decay rate $\left| -\frac{\ln 2}{L_{0.5}-1} \right|$ of CNT quantities as length increases, which results in the faster 'death' of m-CNTs prior to s-CNTs during natural elongating growth. Then, almost all the m-CNTs will decay at a length of 154 mm, leaving the target 99.9999% s-CNTs (Fig. 1a). Careful Raman G mode analysis was conducted by collecting the spectra of ~10$^4$ individual CNTs (Supplementary Fig. 4), where each spectrum was proven to be clearly sensitive to the CNT's chiral structure (Supplementary Fig. 5). All the spectra displayed neither BWF nor D band signals, directly testifying the ultrapure s-CNTs (at least 99.99%). Electron-donating surfactant treatment, which was verified efficient in strengthening BWF[15], also confirmed the enrichment of s-CNTs longer than 154 mm (Supplementary Fig. 6). The aligned CNTs synthesized at other conditions (Supplementary Fig. 7) also demonstrated the ultrahigh s-purity at length >154 mm. It is significant to note that when discussing the rate-selected growth, our measurements and statistics are not sensitive towards any kind of CNTs, single-walled (SWNTs), double-walled (DWNTs) or triple-walled (TWNTs). While there might be a potential connection with the wall numbers that require further study. Although within a narrow range of diameter, the ultralong CNTs could resonate with the laser irradiation wavelength we used, the actual number of the G mode

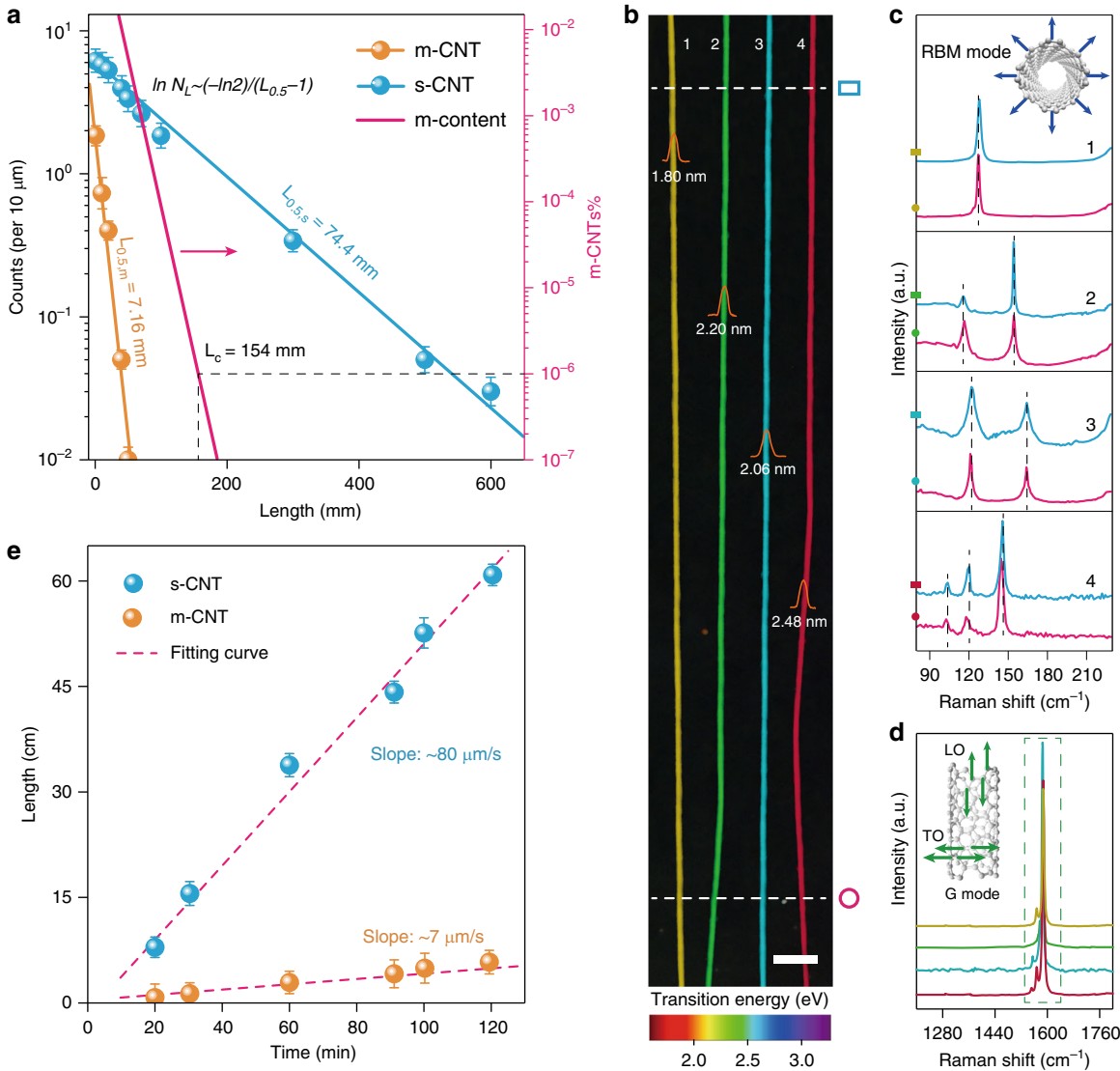

**Fig. 1** Growth behavior analysis of m- and s-CNTs. **a** Quantity statistics of m- and s-CNTs at the same length, by summing up all four samples (~710 CNTs). The m-percentage line is derived from the number of total CNTs divided by that of m-CNTs. **b** Spatial Rayleigh images of representative ultralong CNTs spliced onto the same background. Their respective height traces measured with the atomic force microscope (AFM) are shown inset. Scale bar, 10 μm. **c** RBM modes of the top and bottom parts probed from each tube in **b**. **d** G mode waterfall of the corresponding ultralong CNTs. **e** Growth rates of m- and s-CNTs on array scales. Error bars represent the standard deviation of $n = 10$ longest m- or s-tubes grown at varied growth time

components that can be observed depends on the laser power, chiral structures of CNTs, etc[16]. It cannot permit to identify the wall numbers directly from the splitting G modes even if the metallic components can be sensitively responded in the form of BWF peaks.

**Electrical properties of the rate-selected ultralong s-CNT arrays.** For further verification of the selectivity, we measured the electrical properties of the CNTs longer than 154 mm. These ultralong CNTs were directly synthesized on Si/SiO₂ wafers (Fig. 2a–c) within a layered rectangular reactor, which was designed by us for scaling-up ultralong CNTs with a more uniform laminar distribution[17]. The longest CNT possesses a consistent chirality over a 650-mm-long distance with a high growth rate of 88 μm s⁻¹ (Supplementary Fig. 8), updating a new length record (the highest 550 mm ever reported)[10] of CNTs. Even though the CNTs were densified by multiple growths, transistors fabricated on those tubes longer than 154 mm all demonstrated high on/off ratio after each cycle (Supplementary Fig. 9).

Prototypical ~100 parallel transistors (Fig. 2d–g) were fabricated on aligned long tubes contacted with interdigitated electrodes. At a drain-source bias ($V_{DS}$) of -0.1 V, the device delivered a high width-normalized current of ~14 μA μm⁻¹ for a single channel (corresponding to ~42 mA output for the whole device) with an on/off ratio of 10⁸ and field effect mobility[18] >4000 cm² V⁻¹ s⁻¹ (Fig. 2h). There is no obvious current leakage from the gate while the device is being operated. The impact of the gate current can be neglected on the areal on/off ratio. And the transfer characteristic has been characterized when there are no CNTs working in the channel (Supplementary Fig. 10). The drain voltage of −0.1 V is a common safe parameter for testing CNTs as well as an optimal working condition with less switching power. Although few walled ultralong CNTs with smaller outermost bandgap were reported to carry the higher saturated current than that of SWNTs[12,17], the potential limit of the operating $V_{DS}$ corresponding to the CNT bandgap requires further study.

Covering 1.2-mm-long CNTs, this interdigitated device can still retain high on/off ratio and mobility, which effectively

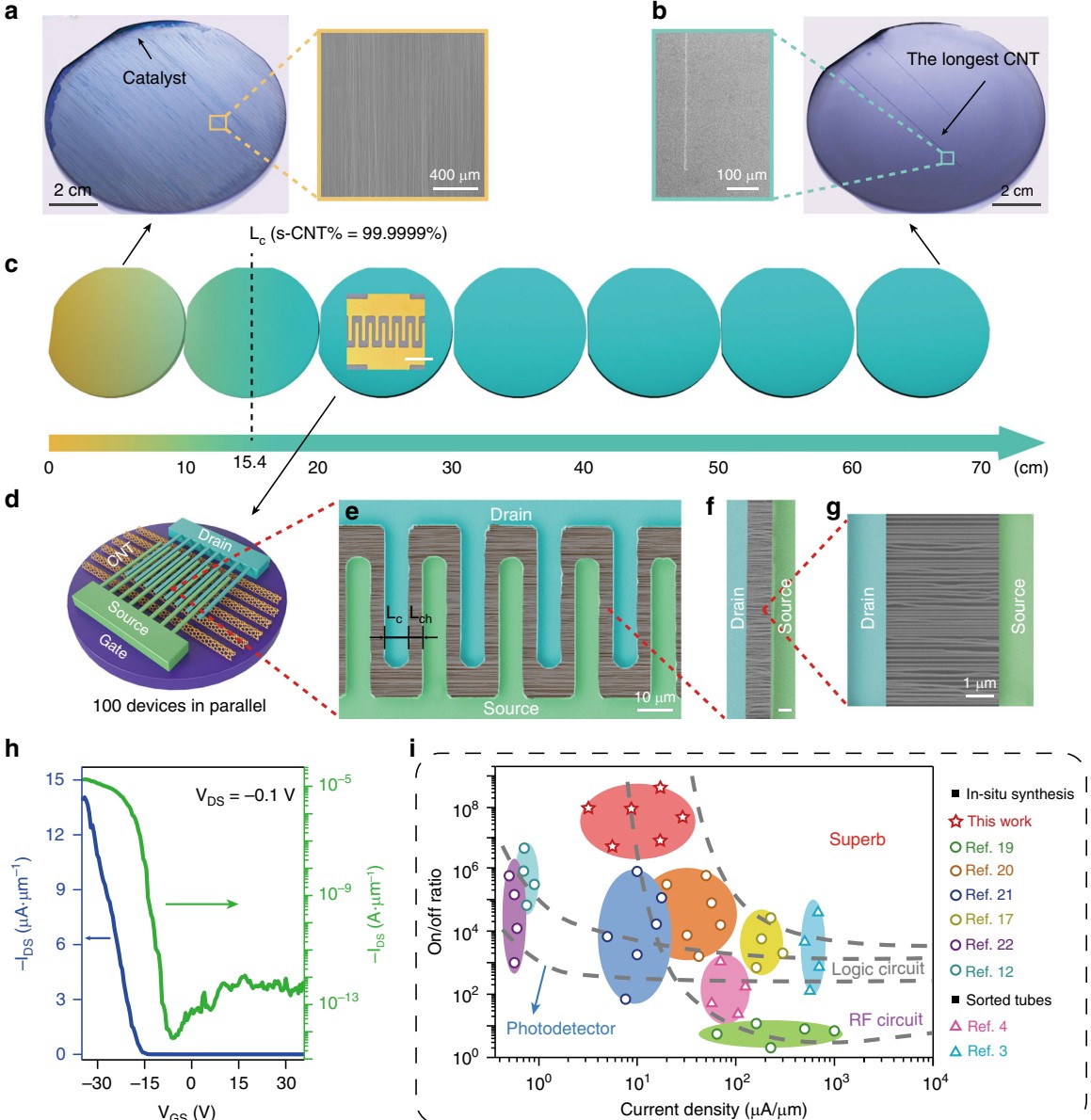

**Fig. 2** Device fabrication with highly pure s-CNTs. **a, b** Vapor-condensation-assisted optical microscopy[38] for the CNTs synthesized on sequentially placed Si/SiO$_2$ wafers (diameters ~100 mm), 1st for (**a**) and 7th for (**b**). Beside panels are expanded scanning electron microscope (SEM) views from the rectangular areas. **c** Schematic depicting the obtainment of ultrapure aligned s-CNTs. Tubes on the left wafer in yellow contain more metallic components, whose content is indicated by the yellow brightness. At the critical length $L_c$ = 154 mm, 99.9999% pure s-CNTs can be obtained. Inset, the optical image of the interdigitated device. Scale bar, 40 μm. **d–g** A prototypical interdigitated transistor fabricated on ultrapure s-CNTs, with a nominal channel length ($L_{ch}$) of 4 μm and contact length ($L_c$) of 8 μm. Explicit device structures are magnified in **e–g**. Scale bar, 2 μm in (**f**). **h** Transfer characteristics of the width-normalized transistor for a single channel plotted in both linear (blue, left axis) and logarithmic (green, right axis) scales with applied $V_{DS}$ of -0.1 V. **i** Benchmarking the width-normalized device fabricated on ultralong CNT arrays for a single channel with those previously reported. The standards for various applications are derived from the proposal of IBM[7], by transforming the target parameters (density and purity of s-CNTs) into measurable electrical quantities with an assumption of the saturated on-current being 10 μA per tube[17]

demonstrates the stable semiconducting property and perfect structure of the long CNTs. However, in a transistor with an interdigitated set of electrodes, the capacitance will increase with the number of fingers so that there is no real delivery gain regarding the use of interdigitated electrodes. The higher current available from the interdigitated electrodes is useful for exploring the limits of CNT performance but not for the practical devices. Therefore, we normalized the performances of the ultralong-CNT-array transistors for the whole length of the channel and compared them with those previously reported[3,4,17,19–22] in

Fig. 2i. It shows a significant improvement of the s-CNT purity. The superior performance of the device based on aligned CNTs comes from the exceptional electrical properties of individual ultralong CNTs. We measured 452 transistors each built on a single CNT with the length over 154 mm (Supplementary Figs. 11 and 12). On average, they have exhibited excellent electrical performances, such as the on/off ratio around $10^6$, on-state conductance of 16 μS and mobility of 4451 cm$^2$ V$^{-1}$ s$^{-1}$. However, it's still hard to understand such high on/off ratio for transistors fabricated on the few-walled CNTs with smaller

bandgap (~0.2 eV). The on current, ~$6 \times 10^7$ holes per cm, is estimated to be the product of $V_{GS}$ and capacitance while the off current is proportional to the thermally generated carrier density[23]. Theoretically, the on/off ratio should be lower than $6 \times 10^3$ for a CNT with the outermost bandgap less than 0.26 eV, which is $10^5$ times lower than that we actually measured. Direct molecular dynamics or Monte Carlo simulations also failed to analyze this anomaly due to the difficulties in simulating non-defective CNT structures of a well-defined diameter and recognizable chirality[24,25]. It seems that an explicit physical theory to bridge the gap between atomistic dynamics and macroscopic scales is required to interpret experiments and simulations.

**The interlocking mechanism for rate-selected growth**. This kind of long-range perfect hexagonal carbon lattice comes from an interlocking between the electronic structures and kinetic properties of the ultralong CNTs. Generally, for solid catalysts, concurrent factors of kinetic and thermodynamic aspects of CNT growth determine the final chirality distribution[24,26–28]. When the metal of liquid catalyst adapts to the CNT edge, there is no energy cost to create a pair of kinks due to their highly mobile structures. Therefore, it can hardly lead to any thermodynamic preference and generally the kinetic route of selection dominates. By neglecting the nascent nucleus formation, we focused on the catalyst combined with infant tube as the new 'template' for subsequent elongating growth. This infant tube acts as a modification to the catalyst, making its catalytic performance vibrant by changing the electronic structures. Overall, catalysts adjacent to semiconducting infant tubes have exhibited higher reactivity than those adjacent to metallic ones, as s-CNTs have an advantage of ten-time kinetic growth rate over m-CNTs. Microscopically, we can redefine the growth rate as turnover frequency (TOF, $s^{-1}$), that the counts of C2 adsorbing onto one circumferential active site per second. $^{13}C/^{12}C$ isotope switching method[29] was used to measure the TOFs of individual CNTs (Supplementary Fig. 13). $^{13}CH_4$ was switched in followed by a duration of conventional $^{12}CH_4$ cracking to produce $^{12}C$-$^{13}C$ heterogeneous tubes. Spot-by-spot micro-Raman measurement was performed to probe the $^{12}C$–$^{13}C$ transition site and calculate the TOF. The equivalent TOF of $^{12}CNT$ and $^{13}CNT$ along individual tubes specified no significant effects of isotope on CNT growth. Just as the Brønsted–Evans–Polanyi relationship in the transition metal catalysis[30], a similar volcano dependence of TOF on bandgap is settled as shown in Fig. 3a (the original data can be seen in Supplementary Fig. 14 and Supplementary Tables 1 and 2). The difference is that the bandgap structure represents the adsorption energy (equals to reaction energy for diatomic molecules) in the CNT catalysis. S-CNTs define a much broader and higher space than m-CNTs versus the bandgaps. The average TOF of s-CNTs (~$1.5 \times 10^6$ $s^{-1}$) is an order of magnitude higher than that of m-CNTs (~$1.3 \times 10^5$ $s^{-1}$) and also the highest among the reported industrial catalytic reactions ($10^{-2}$~$10^2$ $s^{-1}$ for most relevant industrial applications[31,32]). Besides, the fact that the numbers of m-CNTs (or s-CNTs) are identical between the results of Raman spectra and electron diffraction, proves the assumption that these long tubes just resonated with the laser irradiation wavelength we used (Supplementary Fig. 15). Due to the inverse relation between bandgap and diameter, diameter distribution centered around $2.4 \pm 0.3$ nm substantially predominates for those s-CNTs with higher TOF (Fig. 3b). However, no SWNTs have been found in a wide range of diameters among tubes longer than 154 mm. It is consistent with our previous results that SWNTs covered less than 10% while the few-walled ones were kinetically favorable if trace amount of vapor was

added[12]. It seems more challenging to synthesize SWNTs with length up to meter scale possibly due to their higher curvature energy[33]. We assume that the recipe of catalysts capable of strengthening the CNT-metal adhesion may play the key role in facilitating the expansion of the nanotube ends.

## Discussion
Apart from the catalytic effect, the new template also plays the 'affinity' role to confine the orientation of subsequent C2 addition. Although both the survival probability and TOF of the infant tube are discrete when adding C2, their coordination gives rise to a confirmed CNT bandgap and chiral structure. As the growth is kinetics-controlled, once the bandgap is initially locked, consecutive C2 will incorporate onto the CNT at a unique TOF. Any net circumferential disclination caused by topological defects into the hexagonal bond network will distort the van Hove singularities and lead to hybridized bandgaps[8]. This will reasonably induce unstable growth rates along individual tubes and break the SF distribution for populations. Thus, a strong interlocking between bandgap structure and C2 additions provides a potential edge-controlled quasi topology protection, directing the synthesis of atomically perfect m- or s-CNTs. Only ultralong s-CNTs with higher TOF and stable growth rates can survive in the final rate-competitive growth. This high-speed interlocking growth mode can even support $3.9 \times 10^{10}$ steps of dimer-additions with equal survival probabilities and uniform time interval. We anticipate that similar behavior is adaptable for other substances agreeing with the SF distribution. Further explorations are required in the dependence of nanotube growth on their microscopic structures, but this mechanism has already guided a robust and scalable methodology to obtain highly pure s-CNTs with a narrow diameter distribution.

## Methods
**Synthesis of ultralong CNTs in the tube reactor**. The catalysts were prepared according to the preloading strategy[34]. Briefly, 0.7 g catalyst precursor (FeCl$_3$·$n$H$_2$O) powder was heated from room temperature to 900 °C in a continuous H$_2$/Ar ($V_{H2} : V_{Ar} = 2:1$; $F_{total} = 150$ sccm) flow. After the sublimated FeCl$_3$ being reduced by H$_2$, iron catalyst nanoparticles would be embedded onto the inner surface of the quartz reactor once the temperature was cooled down. Then, these surface-anchored catalysts would gradually release under H$_2$ reduction during the growth process, followed by migrating to the target substrate surface to prompt the growth of aligned CNTs. This strategy has been testified efficient in more than 50 times repeated growth of ultralong CNTs, via only a one-time preloading process. The CNT growth process was similar to the method described in our previous publications[10,12,17]. The temperature was increased at a rate of 30–35 °C min$^{-1}$ to 1020 °C and kept for 15 min under H$_2$/Ar atmosphere. Then methane was supplied for the synthesis of ultralong CNTs with hydrogen being used to modulate the reaction balance ($V_{H2} : V_{CH4} = 2:1$; $F_{total} = 75$ sccm, with 0.5% H$_2$O). After 10–120 min growth, the ultralong CNT sample was cooled down to room temperature in the H$_2$/Ar atmosphere ($V_{H2} : V_{Ar} = 2:1$; $F_{total} = 150$ sccm) to protect against the oxygen etching.

**Synthesis of wafer-scale ultralong CNTs in the layered rectangular reactor**. The process of wafer-scale growth of ultralong CNTs in the muffle furnace is similar to the above descriptions. Sequentially placed silicon wafers with a diameter of 10 cm were loaded on a quartz plate and encapsulated into our home-made reactor[17]. The catalysts were deposited with the above-mentioned preloading strategy. Besides, the micro-contact printing strategy was utilized as well to improve the mono-dispersity of catalysts further[35]. In brief, poly(dimethylsiloxane) (PDMS) stamp without ozone oxidation was wet with an ethanol solution of FeCl$_3$·$n$H$_2$O (0.03 M). After evaporation of ethanol, Fe$_2$O$_3$·$n$H$_2$O nanoparticles were formed from hydrolysis. Then these iron oxide nanoparticles were transferred onto the half circumference of the first silicon wafer and acted as the catalysts after being reduced by hydrogen. The CNT growth process resembled the above statement except that the heating rate was 20 °C min$^{-1}$ and the gas composition was $V_{H2}$: $V_{CH4} = 2:1$; $F_{total} = 154$ sccm, with 0.6% H$_2$O content.

**Raman measurement on individual suspended ultralong CNTs**. Raman spectra were collected with Horiba HR 800 equipped with a liquid-nitrogen-cooled silicon CCD detector and three excitation sources at 532 nm (2.33 eV), 633 nm (1.96 eV) and 785 nm (1.58 eV). The scattered light was collected through a 100× air

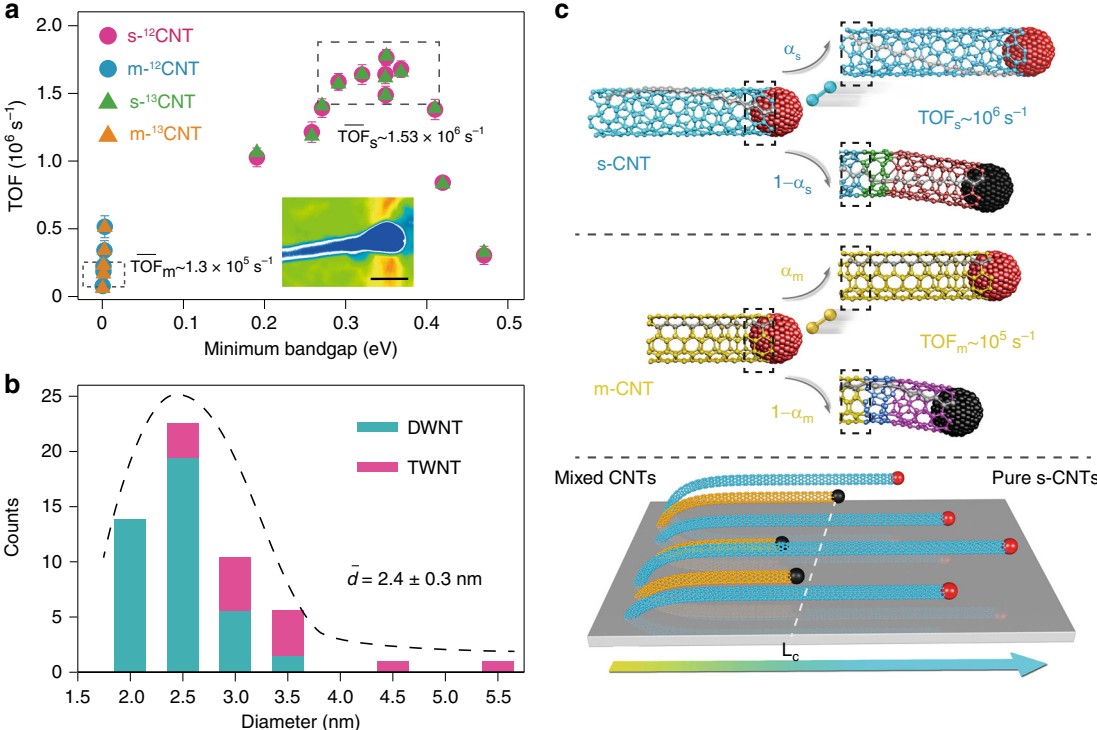

**Fig. 3** Bandgap dependent dimer addition process. **a** Minimum bandgap (target layer) of concentric layers for few-walled CNTs versus TOF of the target layer. Detection of $^{12}C$ preceded $^{13}C$ signal verified the expected tip mode, also indicated by the AFM image on the tip of a long CNT (inset, scale bar, 10 nm). The errors come from the uncertainty of identifying the transition site. **b** Diameter distribution of CNTs longer than 154 mm. This reveals a trend that DWNTs are highly enriched (75%) among long species with the diameter smaller than 3 nm, whereas TWNTs dominate for tube diameter larger than 3.5 nm. **c** Schematic of the TOF differences between m- and s-CNTs. After adding each dimer, both the m- and s-CNT will survive or die, with equal $\alpha$ but different dimer-addition frequencies, which results in differences between $\alpha_m$ and $\alpha_s$. This enables a spontaneous purification of s-CNTs from their m-counterparts

objective (laser spot diameter ~1 μm) using a backscattering configuration. Stokes Raman spectra were acquired with edge filters cutting at $100\,cm^{-1}$. A 600 grooves per mm grating rate was used, giving a spectral resolution better than $2\,cm^{-1}$. Laser power was kept at or below ~1 mW to avoid the heating effect. Typical accumulation time was 60 s for each spectrum. All Raman spectra were corrected by subtracting the background signal collected at a blank slit. The substrates used in our study are silicon wafers with 300-nm-thick thermal oxide on the surface, containing slits (300 nm deep, 5-20 μm wide) and marks fabricated by photo-lithography and dry etching. Micro-Raman measurements were conducted on the suspended CNTs across the slits, manifesting their intrinsic properties with the van der Waals CNT-$SiO_2$ interaction being eliminated[36].

The sample was encapsulated in a stage (THMS600E, Linkam Scientific Instrum., the U.K.) for measurements, preceded an annealing process carried out at 450 °C for 15 min under 100 sccm Ar flow, which helped to remove the adsorbed $O_2$ molecules on the surface of CNTs[37]. In the process of measurement, determine the approximate position of the target CNT under the microscope first. Then, conduct a Raman line mapping perpendicular to the alignment of the CNTs to confirm the specific coordinate of the suspended CNT. After that, shift the laser spot to that position and collect the spectrum with prolonged exposure time to enhance the areal intensity.

**Vapor-condensation-assisted visualization of wafer-scale ultralong CNTs**. A commercial humidifier was reformed with a metal joint and a hosepipe to produce oriented vapor[17]. In case of the vapor flowing at a high speed, Ar gas flow was introduced to the vapor for dilution. Under the light illumination, the vapor was blown to the surface of the Si substrate and the CNTs would be visible with naked eyes[38]. A bottle of liquid nitrogen would be better to put under the substrates to maintain the topography of ultralong tubes so as to be captured with a micro lens.

**Other characterizations of ultralong CNTs**. The as-grown ultralong CNTs were inspected with SEM (JEM 7401F, 1.0 kV), TEM (JEM 2010, 120.0 kV), AFM (Asylum Cypher) to characterize the morphology and structure. An optical microscope (long working distance metallography microscope, FS 70Z) and supercontinuum laser were used for resonant Rayleigh scatting.

**Dependence of CNT quantity $N_L$ on the length $L$**. According to the SF distribution,

$$N_L = N_0 \propto^{L-1} \qquad (2)$$

Here, $L$ denotes the distance from the starting position of a substrate, which is the measured length with a unit length of 1 mm. $N_L$ indicates the total CNT quantity at position $L$.

When the $N_L$ decreases to half of $N_0$,

$$0.5N_0 = N_0 \propto^{L_{0.5}-1} \qquad (3)$$

Then,

$$ln \propto = -\frac{ln2}{L_{0.5}-1} \qquad (4)$$

Logarithm on both sides of Eq. (2),

$$\ln N_L = \ln N_0 + (L-1)\ln \propto \qquad (5)$$

Substitute Eq. (4) into Eq. (5), we can get

$$\ln N_L = \ln N_0 - \frac{ln2}{L_{0.5}-1}(L-1) \qquad (6)$$

**Evaluation on the dimer addition times of the 650-mm-long CNT**. For a 1-nm-long single-walled CNT with a diameter of 1 nm, there will be 60 carbon dimers in the CNT. Therefore, for the 650-mm-long CNT, there will be at least $3.9 \times 10^{10}$ times of dimer-additions with an assumption of the CNT being single-walled with a diameter of 1 nm.

**Fabrication and measurement of the transistors**. Ultralong CNTs were grown on a heavily n-doped silicon wafer with an 800-nm $SiO_2$ top layer. After annealing treatments at Ar atmosphere (450 °C), the adsorbed amorphous carbonaceous impurities were removed, Pd film of 70 nm was then patterned on the wafer using electron beam lithography and deposited via electron beam evaporation under high vacuum. By cutting off the neighboring CNTs, selected aligned long tubes were contacted with Ti/Au (5/40 nm) contacts deposited as the source-drain metal electrodes with a similar process as Pd. Electronic measurements were carried out by applying drain and gate voltages relative to the source electrode with a Keithley

4200 A parameter analyzer at room temperature in air. The drain current was measured with a Keithley 4200-PA amplifier. The low current measurement capabilities of any SMU can be extended by adding an optional 4200-PA preamplifier. The preamplifier provides 10 aA resolution by effectively adding five current ranges to either SMU model.

Field effect mobility ($\mu_e$) of an individual CNT can be calculated according to the equation[18] $\mu_e = \frac{L_{ch}}{C} \frac{dG_{on}}{dV_{gs}}$, here $C$ is the single-tube capacitance. The total single-tube capacitance can be viewed as the harmonic mean of the electrostatic gate capacitance and the quantum capacitance, described by the equation[39] $C_g^{-1} = C_{g,el}^{-1} + C_q^{-1}$. The electrostatic gate capacitance is $C_{g,el} \approx 3.1 \times 10^{-11}$ F m$^{-1}$, significantly smaller than the quantum capacitance $C_q \approx 4 \times 10^{-10}$ F m$^{-1}$ and the $C_q$ term can be ignored. Then the single-tube mobility can be obtained based on the transfer characteristics of the transistor and its device structural parameters.

**Growth rate measurements with the isotope switching tests**. The process of synthesizing the $^{12}$C–$^{13}$C heterogeneous tubes is similar to the above-stated procedure for ultralong CNTs, with $T = 1020$ °C, $V_{H2}$: $V_{CH4} = 2.06$: 1, $W_{H2O} = 0.46\%$. And the carbon feedstocks are $^{12}$CH$_4$ and $^{13}$CH$_4$, with $^{13}$CH$_4$ (99% purity) purchased from Spectra Gases Inc., America. Their pipelines were parallel connected in front of the stable-flow valve and stable-pressure valve, followed by through a three-way valve, in order to achieve a stable gas-switching between $^{12}$CH$_4$ and $^{13}$CH$_4$. The growth process is depicted below.

After synthesis, spot-by-spot micro-Raman measurements were performed along each ultralong CNT of the array. The longest ($x_{b,1} = 25.2$ mm) and a shorter ($x_{b,2} = 1.8$ mm) CNT were specifically analyzed as below. Due to the sensitivity of Raman spectra on the mass change of carbon atoms for both G and RBM modes, an obvious down-shift has been discovered on $^{13}$C nanotubes compared with those composed of $^{12}$C, with a uniform ratio $\omega_{13C} : \omega_{12C} \approx 0.96$. Once the $^{12}$C–$^{13}$C conversion site ($x_a$) is confirmed through the frequency shift, growth rates of $^{12}$CNT ($R_{12C} = x_a / t_a$) and $^{13}$CNT ($R_{13C} = (x_b - x_a)/(t_b - t_a)$) can be calculated, respectively.

The following calculations show that the growth rates of $^{12}$CNT and $^{13}$CNT are the same along an individual ultralong CNT. For the long s-CNT in the above schematic, the measured conversion site is $x_a = 15$ mm, so the growth rate of $^{12}$CNT is $R_{12C} = 83.3$ μm s$^{-1}$ and the growth rate of $^{13}$CNT is $R_{13C} = 83.3$ μm s$^{-1}$, as well. For the short m-CNT, the measured conversion site is $x_a = 1.08$ mm, so the growth rate of $^{12}$CNT is $R_{12C} = x_a / t_a = 6$ μm s$^{-1}$ and the growth rate of $^{13}$CNT is $R_{13C} = 6$ μm s$^{-1}$, as well.

**Measurement of TOF from the growth rates**. The number of carbon atoms (Q_atom) within a 1-nm-long target layer was simulated with the Nanotube Modeler Software. The quantity of active sites for the target layer with chiral index $(n, m)$ has been demonstrated as $m$[40]. The growth rates ($R$) are considered the same for all concentric layers, which were measured with the above isotope switching method[29]. Then the TOF can be expressed as TOF = $R \cdot Q_{atom}/m$.

**Measurement of the CNT bandgap**. For few-walled s-CNTs, the layer with the minimum bandgap often belongs to the outermost one according to the inverse relationship between diameter and bandgap of CNTs. Whereas, if there are any metallic layers within a few-walled s-CNT, the minimum bandgap will turn to the metallic ones. Therefore, in order to precisely obtain the minimum bandgaps of concentric layers, all the as-grown ultralong CNTs were transferred onto the Si/SiO$_2$ grids assisted with cellulose acetate films[12] after the isotope switching tests. The cellulose acetate films were 60 μm in thickness, purchased from Beijing XXBR Tech. Co., Ltd., China. Careful electron diffraction analysis helps to identify the chiral indices of all few-walled CNTs. Then the diameters and conductive properties of concentric layers can be confirmed separately. For semiconducting layers, the bandgaps $\Delta E$ were confirmed from the inverse relationship $\Delta E \sim r^{-1}$ ($r$, the radius of the CNT)[41]. For those metallic layers ($n - m = 3k$, $n \neq m$, k is an integer), the bandgaps were usually not zero due to the presence of curvature derived gaps, which follows the relationship $\Delta E \sim r^{-2}$ [42].

## Data availability

The authors declare that all relevant data supporting the findings of this study are available within the paper and its Supplementary Information files. Additional data are available from the corresponding author upon reasonable request.

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

## Acknowledgements

We are grateful to D. Zhang (Peking University, China), Y. Hu (Tsinghua University, China), L. Wang (Tsinghua University, China) for discussions. This work was supported by the Ministry of Science and Technology of China (2016YFA0200101 and 2016YFA0200102) and the National Natural Science Foundation of China (grant 21636005 and 51872156).

## Author contributions

F.W., J.X., and Z.Z. conceived and designed the experiments. Z.Z., N.W., W.C., S.S., and J.G. performed the experiments. Q.W. and S.S. were responsible for the technical assistance with the electron diffraction pattern of carbon nanotubes using HRTEM. Z.Z., N.W., W.C., B.S., Q.W., S.S., and J.G. performed the data analysis. Z.Z., N.W., R. Z., and Y.W. wrote the paper. All authors discussed the results and commented on the paper.

## Competing interests

The authors declare no competing interests.
