## [Peer Review File · Nature Communications]

Reviewers' Comments:

Reviewer #1:

Remarks to the Author:

This manuscript reports the synthesis of ultra-long and high purity semiconducting CNTs. Since the obtaining high purity semiconducting CNTs is important issue for industrial application of CNTs, the reported results, synthesis of 650 nm long CNTs with high purity, is of interest and can give the strong impact to the variety of scientific communities.

However there are several uncertain points about the results, which should be carefully concerned before the acceptance of this manuscript.

(1) In the abstract, the authors insist the purity of s-CNT is 99.9999%. In the main text the purity is shown as at least >99.99% measured by different method. Which value is reasonable? Since the value of s-purity is one of the key results in this manuscript, much more concrete and clear evidence is required to support the accuracy of this purity estimation. UV-Vis-NIR spectroscopy can also help to discuss the purity of s-CNTs.

(2) In Fig. 1d, G- peaks (splitting G-band) are shown in all CNTs. The G- peak should be shown only for small diameter CNT such as SWNTs. Similar G-band splitting is also shown in Fig. S11. But it is not shown for other Raman spectra such as Fig. S3, S4, S5, S6, and S7. Do SWNTs mix in the samples?

(3) Are there any correlations between inner tube and outer tube for long DWNTs? For example, chirality or diameter of inner and outer tube pairs have some correlation or completely random? Since Table S1 is shown as an experimental result, such kind of discussion should be added in the main text.

(4) The effects of SWNTs have not been discussed in this manuscript.

How about the concentration of SWNTs within total CNTs?

Can similar explanations (s, m selectivity) be applied for SWNTs?

If not why?

These should be discussed in the main text.

Reviewer #2:

Remarks to the Author:

This paper describes selective growth of semi-conducting ultralong carbon nanotubes (CNTs) using a well-established "kite growth" process, explored extensively, particularly by this group, over many years (see their review reference [16]). The growth Schulz-Flory growth kinetics have been discussed previously, including the effects on chirality. The authors have also made previous suggestions about the persistence of semi-conducting CNTs (see section 5, [16]).

The current paper builds on this previous work, measuring a larger number of CNTs, and confirming the expected fast growth of semiconducting CNTs. The kinetics are explored using an established isotopic labelling approach. This tendency is predicted by the theory of Yakobsen, and has been evidenced widely, for example by the detailed studies of Maruyama. Here, there is a new hypothesis that the growth rate is related to band-gap modulation of the catalyst. It's an interesting idea but not well evidenced, and there is no discussion of the usual Yakobsen model. There is a large amount of difficult, detailed experimentation to determine the CNT chiralities by electron diffraction and correlate to the growth rate (though the methodology for maintaining this correlation could be clearer).

However, the chirality distributions are not compared to the Yakobsen model. In addition, the band gap model relies on the lowest band gap, regardless of which shell of the (multiwall) CNT it lies.

Further, the band gap at the discontinuous end of the CNT next to the catalyst is assumed to be the same as that of an infinite CNT which seems unlikely. There are no models of charge transfer or other effects that could explain the dependence.

More practically, by repeated iteration, the authors grow relatively dense arrays, which carry a relatively high current density, and hence in the best case give a high on/off ratio. However, it appears that the high ratio is attributed only to the improved measurement ability at higher currents, rather than a fundamental improvement. It might be more interesting to plot on/off ratio versus mobility rather than current density (as outlined in the review or Rouhi et al), but the devices might look less exceptional (though still very good).

Another major claim is that there is a "spontaneous purification of 99.9999% s-CNTs" is not at all adequately evidenced. It is an extrapolation of the calculated rates, but is not actually shown to apply in practice. There is an attempt to measure 99.99% purity by Raman, but it is not really clear how sensitive the approach is to a small number of m-CNTs. The study is claimed to be a Raman study one by one, but is only given as an approximate number of CNTs (10,000) studied. In general, Raman data is presented without specifying the wavelength used, and the effects of resonance on the resulting data are not discussed. There is clear potential for selection of metallic/semiconducting signals in this way, with some diameter dependence.

It is suggested that the approach provides a 'scalable' approach to s-CNT production, but the absolute number of nanotubes remains low; there is no cheap bulk production here, though useful thin films may be produced. A comparison with other selective vapour growth work (eg from the Kauppinen group) is missing. The implications of producing double or triple wall CNTs rather than single wall CNTs on device performance are not really considered. In general, these CNTs are less desirable, as they have different band gaps in each shell, likely generating a more complex response, likely less responsiveness to gating, and lower absolute band gaps.

Generally, there is a substantial body of work here, which is no doubt of interest to those in the field. However, the relatively few new points of more general interest are not well-evidenced. In places, the argument is hard to follow, due to non-idiomatic language, or non-sequiturs. For example, it is not clear in what sense there is a "seemingly huge Schrodinger's cat state". The title "self-purification" might be more clearly expressed, for example, as "Rate selected growth of pure semi-conducting CNT arrays".

Reviewer #3:

Remarks to the Author:

This manuscript reports observation of strong bandgap-dependent growth rate for few-wall carbon nanotubes, i.e., semiconducting tubes grow as much as 10 times faster than metallic tubes. Taking advantage of this observation, the authors prepared ultralong tubes (> 150 nm) that are predominantly semiconducting tubes and obtained impressive FET device performance. I recommend publication of the paper after the following concerns have been addressed in the revision.

1. The formula describing SF distribution (eq. 1 and eq. 5) contain a quantity (L-1). What is the unit of this quantity? Is L a normalized length instead of the directly measured length in the unit of nm?
2. In Figure 2i, data labelled as "sorted tubes" should be more properly labelled as "in situ synthesis with chirality control" or something to that effect. Device performance of sorted tubes reported in ref 8 should be added to the figure for comparison.

Reviewer #4:

Remarks to the Author:

This was one of the more interesting papers that I have read in a while. The primary achievement is

the demonstration that the growth rate of all-semiconducting few-walled carbon nanotubes is about 10 times faster than few-walled carbon nanotubes containing a metallic nanotube, during the floating synthesis of ultralong (100's of μm) nanotubes. The difference in growth rate insures that sufficiently downstream, predominantly semiconducting nanotubes are found. The authors demonstrate this phenomenon by characterizing the electronic-type of a statistically large body of nanotubes via Raman spectroscopy. TEM diffraction is also used to determine the (n,m) chirality of each wall in roughly 40 nanotubes to more definitely prove that the faster growing nanotubes are all-semiconducting. These are non-trivial syntheses and measurements. Electrical characterization data are also provided. If this synthesis strategy can be improved in the future to enable denser arrays of all-semiconducting nanotubes of smaller diameter (and larger bandgap) then it could enable practical implementations of semiconducting nanotubes in microelectronics.

Based on the reported advances in synthesis and the depth of the characterization, this paper could eventually be publishable in Nature Communications. First, however, two major flaws must be corrected.

1) The reporting of current/width and conductance/width for the interdigitated electrodes is nonsensical and must be removed throughout the abstract, text, and figures. The reason is that these parameters can be increased arbitrarily for any material (whether it be silicon, aligned carbon nanotubes, or random networks of solution processed nanotubes) by using interdigitated electrodes to multiple the effect channel width. The current/width and conductance/width will increase arbitrarily with the number of fingers in the electrodes. Therefore, why not create a million or billion or trillion fingers to skyrocket the claimed current/width or conductance/width towards infinity?

In high-performance integrated circuits, the current per width is an important parameter with respect to the capacitance per width. If the width is held constant then increases in current per width lead to higher performing (faster switching) transistors; thus the desire in industry to increase current per width. However, this assumes the capacitance per width is relatively invariant. In a transistor with an interdigitated set of electrodes, the capacitance will also increase with the number of fingers. Thus, there is no real gain to using interdigitated electrodes. The real channel width increases with the number of fingers (times 2). It is very deceiving to report current/width and conductance/width in interdigitated electrodes without correctly using the real channel width to normalize everything back to reality.

Please report only single-channel conductance (by normalizing by the actual interdigitated width) in: the abstract, manuscript text, Figures 2h,i, and Figures S8,9,10. The mobility and high on/off ratio are already impressive and demonstrate that the nanotubes are primarily semiconducting and of high crystalline quality.

2) The paper uses Raman spectroscopy as a means for gathering extensive statistics on metal versus semiconducting purity of the nanotubes. However, missing from the paper is a Kataura plot or a similar plot that shows that the laser energies used are reasonably close to the spectral resonances of the inner, middle, and outer walls of the range of nanotubes grown. If metallic nanotubes are present in a DWNT or TWNT but are not resonant with the laser then these metallic nanotubes may not show up in the Raman spectrum and the nanotubes may be falsely described as semiconducting types.

Alternatively, if the authors have the Raman spectra of all of the ~ 40 nanotubes that they characterized by TEM diffraction to determine the n,m of each wall, then it should be possible to prove (at least for these 40 nanotubes) that Raman spectroscopy is able to correctly identify all-semiconducting versus metallic-containing nanotubes, with high or 100% accuracy.

Also 3) Please report Drain current versus Gate Voltage characteristics for higher V_D at least up to 1V. Prior work (Zhu et al. Sci. Adv. 2016 2 e1601572) has shown that high on/off ratio can only be obtained at small V_D . If high on/off ratio cannot be obtained at higher V_D for example of 0.5V then the reader should be made aware of this deficiency which may originate due to the small bandgap of the outer nanotubes.

Other important points:

4) The paper states

"The total single-tube capacitance can be viewed as the harmonic mean of the electrostatic gate capacitance and the quantum capacitance, described by the equation $C_{g-1} = C_{g,el-1} + C_{q-1}$. The electrostatic gate capacitance is $C_{g,el} = 7 \times 10^{-9}$ F/m, significantly larger than the quantum capacitance $C_q \approx 4 \times 10^{-10}$ F/m and the $C_{g,el}$ term can be ignored."

However, I calculate that $C_{g,el} = 3 \text{ E-11 F/m}$ (wire-plate separated by 800 nm, dielectric constant 3.9, wire radius = 1.25 nm). Certainly, the C_q should not dominate until the dielectric constant becomes very (< 10 nm) thin.

5) Please comment in the text how it is possible to realize such large on current - off current switching ($> 1E6$ or as high as $1E8$) even though energy gap of outerwall NT is so small (~ 0.2 eV). The on current will be proportional to the V_G times the capacitance = $30V * 3E-11 \text{ F/m} = 6E7$ holes/cm. The off current will be proportional to the thermally generated carrier density. According to Akinwande et al. IEEE TRANSACTIONS ON ELECTRON DEVICES, VOL. 55, NO. 1, JANUARY 2008, this should be about $1E4$ holes/cm for a nanotube with 0.26 eV energy gap at 300K (see Figure 4). The ratio between the on current and off current should be approximately $6E3$ for a 0.26eV nanotube and less for nanotubes with larger outerwalls that have energy gaps less than 0.26eV.

Minor:

6) Fig. 2h. The figure indicates V_{DS} is 0.1V. The caption indicates 1V.

7) Fig. 2e. The scalebar reads 10 mm when probably this should be μm . Same with figure g.

8) Fig. S10e. Units are A but probably they should be μA .

Response to Reviewers' Comments

Reviewer #1

General comments

This manuscript reports the synthesis of ultra-long and high purity semiconducting CNTs. Since the obtaining high purity semiconducting CNTs is important issue for industrial application of CNTs, the reported results, synthesis of 650 nm long CNTs with high purity, is of interest and can give the strong impact to the variety of scientific communities. However there are several uncertain points about the results, which should be carefully concerned before the acceptance of this manuscript.

Reply: Thanks for your positive comments on our work. We'd like to improve our work by following your professional advice.

Comment 1

In the abstract, the authors insist the purity of s-CNT is 99.9999%. In the main text the purity is shown as at least >99.99% measured by different method. Which value is reasonable? Since the value of s-purity is one of the key results in this manuscript, much more concrete and clear evidence is required to support the accuracy of this purity estimation. UV-Vis-NIR spectroscopy can also help to discuss the purity of s-CNTs.

Reply: Thank you for pointing out this important problem. **In fact, rather than assess the specific purity of aligned s-CNTs, we'd like to highlight our strategy to achieve the synthesis of 99.9999% s-CNTs based on Schulz-Flory (SF) distribution.** In our prior work, we demonstrated the exponential law of the quantity of aligned ultralong CNTs changing with length following the SF distribution.¹ Furthermore, by detecting individual CNTs of this work under Raman spectra, we've discovered the respective SF distribution for either m-CNTs or s-CNTs. **If we define the half-length $L_{0.5}$ where the CNT quantity decreases by half compared to that near the catalyst, we have addressed a ten-time difference in $L_{0.5}$ between m- and s-CNTs. Isotope switching tests also verified this notable difference during atomic assembly. From a rational point of view, 99.9999% pure s-CNTs can be obtained at 154 nm through this spontaneously rate selected process.** That is why we're trying to emphasize the achievement of synthesizing 99.9999% pure s-CNTs. In terms of all the other verification

tests, they were just supplements for aligned s-CNTs of such high purity. **Tests were conducted on randomly selected samples no matter for deep Raman analysis or electrical tests.** We hadn't intended to provide the specific purity for s-CNTs as synthesized and it should be another daunting problem in characterization to precisely assess the semiconducting purity when the m-CNTs content approaches the ppm level.

As we know, methods for measuring m-CNTs content include bulk-sample techniques and counting-based techniques. Compared with techniques for bulk samples, counting-based ones can more accurately differentiate between individual m- and s-CNTs based on their electrical and/or optical performances.²⁻⁵ Despite of their tedious fabrication processes and costly instruments, counting-based techniques are more applicable to the sparse CNTs (<1 CNTs/ μm), especially to the horizontally aligned ultralong CNTs. Besides, Raman spectroscopy is common as a way of characterizing individual CNTs owing to its high sensitivity and chiral-selective resonance with laser irradiation wavelength.⁶⁻¹⁰ Particularly, its strong sensitivity of G mode to m-CNT contamination can help quickly identify the conductive properties of CNTs. **Certainly, we'd like to give clearer evidence on the 99.9999% pure s-CNTs longer than 154 μm . But then we must accomplish the huge amount of work to test more than 10^6 tubes one by one. At the same time, even if we finish the whole test, it will still remain a question whether all these CNTs just resonate with the laser irradiation wavelength.** Therefore, apart from testing the Raman spectroscopy of $\sim 10^4$ ultralong CNTs, we treated the samples with surfactants to strengthen the BWF characteristics of G modes, tested tubes longer than 154 μm grown at other conditions and measured the electrical properties of those long CNTs on individual and array scales. **All these supplementary tests were designed and conducted carefully to avoid accidental factors caused by a single test method.**

But still, these supplementary tests are all counting-based techniques. It won't be precise enough unless we make statistics on 10^6 CNTs. **In contrast, UV-Vis-NIR spectroscopy is a kind of bulk sample technique, which can surely help us assess the s-purity. But it will also become ineffective when the m-CNTs content is less than 1%, as the features associated with m-CNT absorption will gradually disappear and subtraction of the dominate background absorption will dramatically influence the calculated results.**¹¹ George *et al.* from the IBM TJ Watson Research Center have realized limits of the UV-Vis-NIR method in precisely quantifying the purity of s-CNTs (>98 to 99%), and turned to counting-based

electrical testing method.¹² Similar electrical testing method has been used as an important means to verify the s-purity in our work. **Additionally, dispersion and tailoring of an individual ultralong CNT during the elution process will cause repeated calculation of a same tube. Especially when m-CNTs are shortened into several parts, we cannot identify the actual m-content in the original sample.** It will be the same with other bulk sample techniques, which demonstrates that this kind of technique may not be quite applicable to the ultralong CNTs.

To summarize, in this work, we'd like to show that this rate selective growth method is a robust and scalable method to achieve the synthesis of target 99.9999% s-CNTs. We have provided enough evidence to demonstrate the difference in half-length $L_{0.5}$ between m-CNTs and s-CNTs, including Raman measurement on 10^4 individual CNTs and hundreds of device detection. **All these experiments were random sampling and directly verified the validity of allometric decay of m-CNTs and s-CNTs following the respective SF distribution. It seems not significant enough to be troubled by the specific quantity we have measured, as on the one hand there have been no convenient assessment methods so far and on the other hand it's not the key point we had intended to highlight.**

Comment 2

In Fig. 1d, G- peaks (splitting G-band) are shown in all CNTs. The G- peak should be shown only for small diameter CNT such as SWNTs. Similar G-band splitting is also shown in Fig. S11. But it is not shown for other Raman spectra such as Fig.S3, S4, S5, S6, and S7. Do SWNTs mix in the samples?

Reply: Thanks for your question. We admit that it's really complicated in the correspondence between G/RBM modes and CNTs, especially for few-walled CNTs, like DWNTs and TWNTs. To better answer this question, we'd like to illustrate first why the G modes are split in SWNTs. As we know, SWNTs are seamless cylinders rolled by a single atomic layer of graphene. In a 2D graphene the G mode shows a single peak feature due to the degeneracy of the in-phase longitudinal optical (iLO) and transverse optical (iTO) phonons.¹³ While in SWNTs, the curvature of graphene will lift this degeneracy, giving rise to a bimodal G-band composed of G^+ and G^- sub features. **Special cases exist in achiral SWNTs (armchair and zigzag SWNTs) where only one component (A_{1g} symmetry) can be observed.**⁹ This demonstrates that the

splitting G-band feature is not available in the Raman spectroscopy of all SWNTs.

In terms of DWNTs, depending on the chiral or achiral character of each constituent SWNT, one expects to observe 4 (chiral @ chiral), 3 (chiral @ achiral or achiral @ chiral) or 2 (achiral @ achiral) components in the G mode spectrum. **However, some components can appear at close frequencies which cannot be experimentally resolved. The areal observed number of components can also be influenced by both laser conditions and CNT structures. All the components will be available when at least one layer of DWNTs resonates with the laser wavelength due to the quantum coupling between layers.** ¹⁴ But the structural characteristics like diameters and interlayer distance will determine the ultimate line shape and peak numbers of DWNTs in the G mode spectrum. D.I. Levshov *et al.* have demonstrated the splitting G mode feature of DWNTs at a proper interlayer distance.⁸ Negative effective pressure caused by interlayer distance larger than 0.34 nm will help to separate the components of the DWNTs. Similar splitting G mode feature (1519, 1574, 1591 cm^{-1}) has also been reported by M.S. Dresselhaus *et al.* in DWNTs of the metallic @ semiconducting configuration.⁷

The principle of splitting G mode in DWNTs also applies to TWNTs. But the interaction among layers becomes more complicated. **It cannot generally visualize all the components if there is only one layer resonating with the laser wavelength.** Inevitable overlapping among peaks will also bring more challenges in distinguishing different components. M.S. Dresselhaus *et al.* resolved experimentally only 3-4 G modes each on five individual TWNTs.¹⁵ After fitting the collective G mode of a TWNT with Lorentzian profiles, D.I. Levshov *et al.* also observed three splitting G modes on a TWNT.⁸ **Therefore, splitting G mode feature is not typical of SWNTs. The observed number of G modes on few-walled CNTs is tightly associated with the resonant condition, chiral structure and their interlayer distance.**

According to our testing results on $\sim 10^4$ CNTs longer than 154 μm , most of these long CNTs are few walled as their RBM modes almost exhibit multiple peaks, which is consistent with the previous results we have reported.^{1,16} TEM images (Fig. S12) and chiral index analysis (Table S1-2) have provided supplementary evidence on these few-walled CNTs. **In fact, G modes in Fig. 1d, Figs. S3-7, Fig. S11 are all multiple peaks with different degrees of splitting.** On the one hand, **single G peak is only for achiral SWNTs, which are really hard to synthesize due to their energetically unstable chiral structures.** On the other hand,

improper interlayer distances in few walled CNTs will give rise to overlapping of G mode components. This can be eliminated by fitting the collective G mode with multiple Lorentzian profiles, just as the method reported by D.I. Levshov.⁸

Comment 3

Are there any correlations between inner tube and outer tube for long DWNTs? For example, chirality or diameter of inner and outer tube pairs have some correlation or completely random? Since Table S1 is shown as an experimental result, such kind of discussion should be added in the main text.

Reply: Thanks for your question. Certainly, it's really important to discuss the correlations between layers for DWNTs although it has been controversial for decades. Early studies on atomic correlation provided a rather constant diameter difference close to 0.75 nm under TEM but no chiral angle correlation was observed.¹⁷ However, more detailed studies have recently demonstrated the quantum coupling between layers from the collective modes in Raman spectroscopy.^{14,18} **The helicities between inner and outer tubes of DWNTs are usually less than 15° in favor of the minimized strain effects.**¹⁹ This kind of dependent orientations of hexagonal carbon network also applies to the ultralong DWNTs as synthesized. But it seems that inner and outer tubes are more strongly correlated in s-CNTs than in m-CNTs, as **85% of the s-CNTs possess twinning angles where $\Delta\theta < 15^\circ$ and the m-CNTs is only 44% vice versa.** Commensurate configurations $\Delta\theta = 0^\circ$ are not energetically favorable and rarely observed in our samples, which has also been verified *via* our previous demonstrations of the extracted inner walls.²⁰ In terms of the long MWNTs (both DWNTs and TWNTs included), we've discovered that **86% of the tube walls possess helicities higher than 10 °, the ratio of semiconducting walls with helicities over 10 ° even exceeds 90%.** It's typical of horizontally aligned ultralong CNTs with a narrower chirality distribution, as the bulk-synthesized agglomerated and vertically aligned CNTs generally possess at least one layer with helicities close to 0°.^{21,22} Besides, **near-armchair chiral types are favored in CNTs grown at high growth rates. 62% of the s-CNTs contain one layer with the helicity higher than 25 °.** It suggests that MWNTs with larger chiral angles are generally stable enough to withstand high-rate and long-range growth. However, although we've discovered these peculiar distributions, further research is still waiting for us to explore and we'd like to conduct deeper

analysis before we report the chirality distribution. We're expecting for your kind understanding.

Comment 4

The effects of SWNTs have not been discussed in this manuscript. How about the concentration of SWNTs within total CNTs? Can similar explanations (s, m selectivity) be applied for SWNTs? If not why? These should be discussed in the main text.

Reply: Thanks for your question. SWNTs are indeed important in both the science of controlled synthesis and applications of optoelectronic devices. We didn't intentionally avoid discussing SWNTs, but actually, **there is no preference towards any kind of CNTs (SWNTs, DWNTs or TWNTs) when analyzing the selectivity.** According to our explanations in Comment 1, **we cannot identify the specific wall numbers just according to G mode, as the laser conditions, CNT structures, overlapping peaks, etc. will all influence the areal observed numbers of components.** RBM modes may provide another supportive evidence to further identify the wall numbers based on their presented peak numbers. However, generally not all the walls of few-walled CNTs will just resonant with the same laser wavelength, giving rise to the problem of unconformity between the wall numbers and the observed components in the areal RBM spectroscopy. Besides, SWNTs with diameters larger than 2 nm are commonly not available in the RBM modes because of their far too low frequency. Therefore, although Raman is utilized as the main means to identify conductive properties of CNTs, it cannot permit to identify the wall numbers with a good accuracy. In contrast, BWF shaped G mode is more sensitive to metallic components no matter for SWNTs or concentric walls of few-walled CNTs. We have paid more attention to the BWF modes of aligned CNTs at different length positions in our characterization without considering the wall numbers. **Therefore, precisely speaking, the explanations on the selectivity should apply to the as-grown ultralong CNTs without distinguishing SWNTs, DWNTs and TWNTs.**

The reason for why we highlighted DWNTs and TWNTs in Fig. 3b is that **we've only found DWNTs and TWNTs among the tubes longer than 154 mm. That is to say, the concentration of SWNTs longer than 154 mm is zero.** In fact, most SWNTs with short length were discovered to concentrate in the catalyst and short tube zones. **Just as our previous work shows, few-walled CNTs with higher growth rates were kinetically favorable if trace amount of vapor was added during growth, while the ratio of SWNTs grown at 1005 °C**

covered only less than 10%.¹⁶ Compared with few-walled CNTs, it seems more challenging to synthesize SWNTs with length up to meter scale possibly due to their higher curvature energy, which commonly leads to weaker SWNT-metal adhesion strength and negative dissociation energy.²³ Although 185 mm long SWNTs were synthesized with Fe-Mo catalysts,²⁴ few SWNTs longer than 100 mm have been found in our as-grown ultralong CNTs. We assume that the recipe of catalysts capable of strengthening the carbon-metal adhesion may play the key role in preventing nanotube closure.

In order to make it clearer, we'd like to make the following revision (the words in green were copied from the main text while the red words are those after revising).

In the main text

The larger $L_{0.5,s}$ will lead to a slower decay rate $\left| -\frac{\ln 2}{L_{0.5} - 1} \right|$ of CNT quantities as length increases, which results in the faster 'death' of m-CNTs prior to s-CNTs during natural elongating growth. Then, almost all the m-CNTs will decay at a length of 154 mm, leaving the target 99.9999% s-CNTs (Fig. 1a). A G mode collective (Supplementary Fig. 4) statistically counted from $\sim 10^4$ tubes longer than 154 mm, displayed neither BWF nor D band signals, directly testifying the ultrapure s-CNTs (at least >99.99%). Electron-donating surfactant treatment, which was verified efficient in strengthening BWF²⁵, further confirmed the enrichment of s-CNTs longer than 154 mm (Supplementary Fig. 5). The aligned CNTs synthesized at other conditions (Supplementary Fig. 6) also demonstrated the ultrahigh s-purity at length >154 mm. It's worthy to note that there is no preference towards any kind of CNTs, single-walled (SWNTs), double-walled (DWNTs) or triple-walled (TWNTs) when discussing the rate selected growth. Despite of that the lasers we used can excite the ultralong CNTs within limited diameter distributions, the laser conditions, CNT structures, overlapping peaks, etc. will all influence the final observed numbers of components.⁸ It cannot permit to identify the wall numbers directly from the splitting G modes although the metallic components can be sensitively responded in the form of BWF peaks.

In the main text

...The average TOF of s-CNTs ($\sim 1.5 \times 10^6 \text{ s}^{-1}$) is an order of magnitude higher than that of

m-CNTs ($\sim 1.3 \times 10^5 \text{ s}^{-1}$) and also the highest among the reported industrial catalytic reactions. Besides, the resonance of these long tubes with the lasers was confirmed again from the correspondence of m-CNT (s-CNT) quantities between BWF (Lorentzian) shaped peaks and chirality identification under TEM (Supplementary Figs. 12-13). Due to the inverse relation between bandgap and diameter, diameter distribution centered around $2.4 \pm 0.3 \text{ nm}$ significantly predominates for those s-CNTs with higher TOF (Fig. 3b). However, no SWNTs have been found in a wide range of diameters among tubes longer than 154 nm. This is consistent with our previous results that SWNTs covered less than 10% while the few-walled ones were kinetically favorable if trace amount of vapor was added.¹⁶ It seems more challenging to synthesize SWNTs with length up to meter scale possibly due to their higher curvature energy.²³ We assume that the recipe of catalysts capable of strengthening the CNT-metal adhesion may play the key role in facilitating expansion of the nanotube end.

Reviewer #2

General comments

This paper describes selective growth of semi-conducting ultralong carbon nanotubes (CNTs) using a well-established “kite growth” process, explored extensively, particularly by this group, over many years (see their review reference [16]). The growth Schulz-Flory growth kinetics have been discussed previously, including the effects on chirality. The authors have also made previous suggestions about the persistence of semi-conducting CNTs (see section 5, [16]).

Reply: Thanks for your attention on our series of work in ultralong CNTs and we appreciate your wonderful comments on these main research points.

Comment 1

The current paper builds on this previous work, measuring a larger number of CNTs, and confirming the expected fast growth of semiconducting CNTs. The kinetics are explored using an established isotopic labelling approach. This tendency is predicted by the theory of Yakobsen, and has been evidenced widely, for example by the detailed studies of Maruyama. Here, there is a new hypothesis that the growth rate is related to band-gap modulation of the catalyst. It's an interesting idea but not well evidenced, and there is no discussion of the usual Yakobsen model.

Reply: Thanks for pointing out this important problem. **The usual Yakobson model focuses on the concurrent factors of kinetic and thermodynamic aspects of CNT growth that results in chirality distribution. Besides, it highlights the chirality-dependent growth rates of CNTs under a paradigm of screw dislocation.**^{26,27} However, these theories are originally based on SWNTs catalyzed under the solid catalyst systems. **In contrast, when the metal of liquid catalyst adapts to the CNT edge, there is no energy cost to create a pair of kinks on an armchair edge due to their irregular and highly mobile structures.** Therefore, it can hardly lead to any thermodynamic preference and generally causes a broaden chirality distribution, while the thermodynamic nucleation barrier on solid catalysts is lower for the kinkless edges. **Although Yakobson has indicated some chirality distribution of CNTs catalyzed with liquid metal, it seems not quite applicable for our ultralong few walled CNTs according to the results shown in Tables S1-2. This is maybe derived from the more complicated interaction among tube walls.**

It's really complex to understand the whole growth process of ultralong CNTs, especially the nucleation stage, which includes addition of hexagonal and pentagonal rings to a nascent nucleus and the elastic interaction with cap. **In our model, we have combined the catalyst with the infant tube as a new catalytic template, as there has been no effect of liquid catalysts on the selectivity of CNTs until the formation of such infant tubes with definite bandgap.** Just as you said, these infant tubes can be considered as a mediation on the catalytic performances with their bandgap. According to the Brønsted-Evans-Polanyi relationship in the transition metal catalysis, appropriate bandgaps may help contribute to improving the activity of catalysts and thus cause higher growth rates of CNTs. **In such cases where there is no thermodynamic preference during nucleation, the kinetic route of selection dominates in the growth of liquid-catalyzed CNTs, which is consistent with the conclusion of the Yakobson model. In fact, previous molecular dynamics or Monte Carlo simulations have demonstrated failure in producing nondestructive CNT structures of a recognizable chirality.** It seems that bandgap is the neglected factor compared with the more concerned CNT-metal interface because of the challenges in maintaining well-defined electronic structures during modelling.

In order to make it clearer, we'd like to make the following revision (the words in green were copied from the main text while the red words are those after revising).

In the main text

This kind of long-range perfect hexagonal carbon lattice comes from an interlocking between the electronic structures and kinetic properties of the ultralong CNTs. **Generally on solid catalysts, concurrent factors of kinetic and thermodynamic aspects of CNT growth determine the final chirality distribution.²⁷⁻²⁹ Whereas, when the metal of liquid catalyst adapts to the CNT edge, there is no energy cost to create a pair of kinks due to their highly mobile structures. Therefore, it can hardly lead to any thermodynamic preference and generally the kinetic route of selection dominates. By neglecting the nascent nucleus formation, we focus on the catalyst combined with infant tube as the new 'template' for subsequent elongating growth....**

Comment 2

There is a large amount of difficult, detailed experimentation to determine the CNT chiralities

by electron diffraction and correlate to the growth rate (though the methodology for maintaining this correlation could be clearer).

Reply: Thanks for your question. Of course, it took us much time and effort to determine the CNT chiralities and correlate them to the growth rates of corresponding CNTs. **Although the detailed process is pretty complicated, these techniques have been well developed since ten years ago and were reported in our previous work.**¹⁶ Therefore, it's not quite difficult to conduct these experiments. Here, we'd like to highlight the key points of the whole process. **Firstly, the growth substrates are marked with numbers, which helps to accurately locate and differentiate among the CNTs. Besides, the substrates are lithographically fabricated with uniformly distributed slits, giving rise to individual long CNTs divided into parts during their stretching over slits.** All these points have been mentioned in the Method section 'Raman measurement on individual suspended ultralong CNTs'. *The substrates used in our study are silicon wafers with 300-nm-thick thermal oxide on the surface, containing slits (300 nm deep, 5-20 μm wide) and marks fabricated by photolithography and dry etching.* **To make it clearer, we have separated the whole process into three technical sub-parts attached in the Methods section.** They are 'Growth rate measurements with the isotope switching tests', 'Measurement of TOF from the growth rates' and 'Measurement of the CNT bandgap'. These techniques contribute to effectively and robustly transferring only parts of the ultralong CNTs onto the grids for further TEM characterization without damaging residual parts of the same CNTs. It's really helpful to establish the correspondence between TOF and chiralities of the same CNTs.

Comment 3

However, the chirality distributions are not compared to the Yakobsen model.

Reply: Thanks for your comment. In this work, we didn't intend to give more insights about the chirality distributions. **We'd rather attract more attention on our strategy to *in-situ* control the s-CNT purity and the interlocking mechanism to achieve structural control in terms of vapor-liquid-solid growth mode.** Early studies on atomic correlation provided a rather constant diameter difference close to 0.75 nm under TEM but no chiral angle correlation was observed.¹⁷ However, more detailed studies have recently demonstrated the quantum coupling between layers from the collective modes in Raman spectroscopy.^{14,18} **The helicities between**

inner and outer tubes of DWNTs are usually less than 15° in favor of the minimized strain effects.¹⁹ This kind of dependent orientations of hexagonal carbon network also applies to the ultralong DWNTs as synthesized. But it seems that inner and outer tubes are more strongly correlated in s-CNTs than in m-CNTs, as 85% of the s-CNTs possess twinning angles where $\Delta\theta < 15^\circ$ and the m-CNTs is only 44% vice versa. Commensurate configurations $\Delta\theta = 0^\circ$ are not energetically favorable and rarely observed in our samples, which has also been verified *via* our previous demonstrations of the extracted inner walls.²⁰ In terms of the long MWNTs (both DWNTs and TWNTs included), we've discovered that 86% of the tube walls possess helicities higher than 10° , the ratio of semiconducting walls with helicities over 10° even exceeds 90%. It's typical of horizontally aligned ultralong CNTs with a narrower chirality distribution, as the bulk-synthesized agglomerated and vertically aligned CNTs generally possess at least one layer with helicities close to 0° .^{21,22} Besides, near-armchair chiral types are favored in CNTs grown at high growth rates. 62% of the s-CNTs contain one layer with the helicity higher than 25° . It suggests that MWNTs with larger chiral angles are generally stable enough to withstand high-rate and long-range growth. However, although we've discovered these peculiar distributions, further research is still waiting for us to explore and we'd like to conduct deeper analysis before we report the chirality distribution. We're expecting for your kind understanding.

Comment 4

In addition, the band gap model relies on the lowest band gap, regardless of which shell of the (multiwall) CNT it lies.

Reply: Thanks for your question. We believe that there should be no problem for all-semiconducting MWNTs, because the outermost layers possess the minimum bandgap due to the inverse relationship between bandgap and diameter of CNTs.³⁰ It has been mentioned in the Method section 'Measurement of the CNT bandgap'. In terms of m-CNTs, we have intended to choose the bandgap of the outermost layer in subsequent discussions. However, it may not be representative for a metallic MWNT because of the random positions in which layer the metallic components locate. In our model, bandgap is the most important factor we care about. If the outermost layer isn't metallic, it won't be able to construct a fair comparison between m-MWNTs and s-MWNTs, as the minimum

bandgap can be another constraint on the outermost layer of s-MWNTs. Besides, actually, we didn't purposely focus on the outermost layer for neither m-MWNTs nor s-MWNTs. Any layer should be equivalent when discussing the dependence of TOF, as all the concentric layers of individual MWNTs are considered with same growth rate. **Additionally, indicated by the lower decay of aligned s-CNTs with length, a hypothesis naturally arises that smaller bandgap may cause lower growth rates.** Therefore, during the subsequent modelling establishment, we assume that the layer with the minimum bandgap should possibly be the potential limited factor for the high-rate growth of MWNTs. It's a relatively strict way to consider the minimum bandgap despite of many other factors that may still influence the growth of MWNTs, such as the bandgap of inner layers or intermediate layers, certain coupling among the electron wave functions, etc.

Comment 5

Further, the band gap at the discontinuous end of the CNT next to the catalyst is assumed to be the same as that of an infinite CNT which seems unlikely. There are no models of charge transfer or other effects that could explain the dependence.

Reply: Thanks for your questions. It's indeed complicated to discuss the discontinuous interface between catalyst and the tube. **However, it's just because of the lack in clear charge transfer model or theories that we're trying to propose this new template model by combining catalyst with the infant tube.** Before we give more insights on our model, we'd like to elaborate on the growth of ultralong CNTs following the tip growth mode. It has been demonstrated that the tip-growth mode follows the "kite-mechanism" with the catalyst nanoparticle having a long nanotube tail floating in the gas flow.³¹ The whole growth process includes the nucleation stage with the formation of infant tubes, elongation stage with the persistent chiral structure and growth termination stage. **Actually, we didn't intend to emphasize the consistent chiral structures through all these stages but the persistency during the longest elongation stage.**

Firstly, the nucleation stage involves high-speed carbon dissolution and precipitation with a vapor-liquid-solid phase transition. It's difficult to maintain perfect infant tubes with definite chiral structures throughout, as such unstable vapor-liquid-solid transition processes don't generally cause chirality selectivity. However, after the formation of infant tubes, they will be lifted under the convection flow caused by temperature difference,

with the catalyst nanoparticles floating continuously on the tip. The carbon dimers will then assemble onto the rim of the infant tubes along definite chiral directions unless the laminar conditions change. **Therefore, such infant tubes on the one hand play the role of templates directing the subsequent assembly of carbon dimers, while on the other hand determine the areal kinetic assembly rate as a mediation on the original catalysts.** As for the termination stage, we don't think it should be included in our range of discussions. Because any CNT is destined to 'die' due to the decreased activity of catalysts or excessive amorphous carbon coating. **We care less about how an ultralong CNT gets dead, although previous studies have indicated that changes of chirality within centimeter scale should be the main reason of causing death.**¹⁶ It's worthy to note that our model is just an idea to help us understand the whole growth process by simplifying the nucleation and termination stages. It doesn't matter to the practical applications as the elongation stage is long enough to satisfy our requirements for high-end electronics.

Comment 6

More practically, by repeated iteration, the authors grow relatively dense arrays, which carry a relatively high current density, and hence in the best case give a high on/off ratio. However, it appears that the high ratio is attributed only to the improved measurement ability at higher currents, rather than a fundamental improvement. It might be more interesting to plot on/off ratio versus mobility rather than current density (as outlined in the review or Rouhi et al), but the devices might look less exceptional (though still very good).

Reply: Thanks for your recognition and comments. In the main text, we have stated that **'unlike the single-tube transistors, much higher on- and off-currents delivered by aligned CNTs can hardly be smeared out by the minimum ~0.1 pA-level noise detection limit'**. **In fact, there is no obvious difference between our analyzer and those utilized in other groups for electrical measurement. What we'd like to emphasize here is that the detected on/off ratio in Fig. 2h is more close to the true value that CNTs should have exhibited.** Because the minimum detection limit is around 0.1 pA level for general analyzers, which will cause a smear-out on the low off-current of individual CNTs. But there will be no such problems for the aligned perfect ultralong CNTs.

In our opinion, it's better to plot on/off ratio versus current density just as shown in Fig.

2i. Firstly, the devices will look much better if we plot on/off ratio versus mobility, as the mobility of our devices is already excellent enough. If we correct it further by subtracting the effect of contact resistance, the mobility can be close to $10^5 \text{ cm}^2/(\text{V}\cdot\text{s})$.²⁴ But generally, it's not accurate enough to discuss the mobility of sparsely aligned CNTs. This will result in considerable errors no matter with the parallel plate model³² or rigorous model³³. **In contrast, it's more practical to plot on/off ratio versus current density. They respectively reflect the switching performance and current delivery capacity of the devices, while the mobility is more of an indicator of the properties of channel materials.** IBM has launched a roadmap for the next generation high-end CNT transistors, elucidating the importance and requirements for CNTs' semiconducting purity and density.³⁴ **On/off ratio and current density are just the corresponding parameters in another form that can be electrically measured.** By plotting on/off ratio versus current density, it can best exhibit the potential of ultralong CNTs in practical applications although we can make the devices look better in another form of exhibition.

Comment 7

Another major claim is that there is a "spontaneous purification of 99.9999% s-CNTs" is not at all adequately evidenced. It is an extrapolation of the calculated rates, but is not actually shown to apply in practice. The study is claimed to be a Raman study one by one, but is only given as an approximate number of CNTs (10,000) studied.

Reply: Thank you for pointing out this important problem. **In fact, rather than assess the specific purity of aligned s-CNTs, we'd like to highlight our strategy to achieve the synthesis of 99.9999% s-CNTs based on Schulz-Flory (SF) distribution.** In our prior work, we demonstrated the exponential law of the quantity of aligned ultralong CNTs changing with length following the SF distribution.¹ Furthermore, by detecting individual CNTs of this work under Raman spectra, we've discovered the respective SF distribution for either m-CNTs or s-CNTs. **If we define the half-length $L_{0.5}$ where the CNT quantity decreases by half compared to that near the catalyst, we have addressed a ten-time difference in $L_{0.5}$ between m- and s-CNTs. Isotope switching tests also verified this notable difference during atomic assembly. From a rational point of view, 99.9999% pure s-CNTs can be obtained at 154 mm through this spontaneously rate selected process.** That is why we're trying to emphasize the achievement of synthesizing 99.9999% pure s-CNTs. In terms of all the other verification

tests, they were just supplements for aligned s-CNTs of such high purity. **Tests were conducted on randomly selected samples no matter for deep Raman analysis or electrical tests.** We hadn't intended to provide the specific purity for s-CNTs as synthesized and it should be another daunting problem in characterization to precisely assess the semiconducting purity when the m-CNTs content approaches the ppm level.

As we know, methods for measuring m-CNTs content include bulk-sample techniques and counting-based techniques. Compared with techniques for bulk samples, counting-based ones can more accurately differentiate between individual m- and s-CNTs based on their electrical and/or optical performances.²⁻⁵ Despite of their tedious fabrication processes and costly instruments, counting-based techniques are more applicable to the sparse CNTs (<1 CNTs/ μm), especially to the horizontally aligned ultralong CNTs. Besides, Raman spectroscopy is common as a way of characterizing individual CNTs owing to its high sensitivity and chiral-selective resonance with laser irradiation wavelength.⁶⁻¹⁰ Particularly, its strong sensitivity of G mode to m-CNT contamination can help quickly identify the conductive properties of CNTs. **Certainly, we'd like to give clearer evidence on the 99.9999% pure s-CNTs longer than 154 μm . But then we must accomplish the huge amount of work to test more than 10^6 tubes one by one. At the same time, even if we finish the whole test, it will still remain a question whether all these CNTs just resonate with the laser irradiation wavelength.** Therefore, apart from testing the Raman spectroscopy of $\sim 10^4$ ultralong CNTs, we treated the samples with surfactants to strengthen the BWF characteristics of G modes, tested tubes longer than 154 μm grown at other conditions and measured the electrical properties of those long CNTs on individual and array scales. **All these supplementary tests were designed and conducted carefully to avoid accidental factors caused by a single test method.**

But still, these supplementary tests are all counting-based techniques. It won't be precise enough unless we make statistics on 10^6 CNTs. In contrast, UV-Vis-NIR spectroscopy is a kind of bulk sample technique, which can surely help us assess the s-purity. **But it will also become ineffective when the m-CNTs content is less than 1%, as the features associated with m-CNT absorption will gradually disappear and subtraction of the dominate background absorption will dramatically influence the calculated results.**¹¹ George *et al.* from the IBM TJ Watson Research Center have realized limits of the UV-Vis-NIR method in precisely quantifying the purity of s-CNTs (>98 to 99%), and turned to counting-based electrical testing

method.¹² Similar electrical testing method has been used as an important means to verify the s-purity in our work. **Additionally, dispersion and tailoring of an individual ultralong CNT during the elution process will cause repeated calculation of a same tube. Especially when m-CNTs are shortened into several parts, we cannot identify the actual m-content in the original sample.** It will be the same with other bulk sample techniques, which demonstrates that this kind of technique may not be quite applicable to the ultralong CNTs.

To summarize, in this work, we'd like to show that this rate selective growth method is a robust and scalable method to achieve the synthesis of target 99.9999% s-CNTs. We have provided enough evidence to demonstrate the difference in half-length $L_{0.5}$ between m-CNTs and s-CNTs, including Raman measurement on 10^4 individual CNTs and hundreds of device detection. **All these experiments were random sampling and directly verified the validity of allometric decay of m-CNTs and s-CNTs following the respective SF distribution. It seems not significant enough to be troubled by the specific quantity we have measured, as on the one hand there have been no convenient assessment methods so far and on the other hand it's not the key point we had intended to highlight.**

Comment 8

There is an attempt to measure 99.99% purity by Raman, but it is not really clear how sensitive the approach is to a small number of m-CNTs.

Reply: Thanks for your comments. Due to the phono-wave vector confinement and symmetry-breaking effects associated with the curved structure, G mode of SWNTs generally exhibits multi-peaks, including a G^+ feature at 1590 cm^{-1} and a diameter-dependent G^- feature at 1570 cm^{-1} .⁶ **The broad and asymmetric Breit-Wigner-Fano (BWF) line shape of the G mode has long been recognized as a hallmark to distinguish m-CNTs from s-CNTs.**^{10,28,35,36} There is a strong electron-phonon coupling called Kohn anomaly that originates from the interference between the discrete G mode and the continuous electronic Raman scatterings. This gives rise to the broadened and softened BWF line shape of m-CNTs, obviously different from the Lorentzian one of s-CNTs.

To improve the sensitivity of BWF line shapes towards m-CNT contents, we carefully designed and conducted the following demonstrations which have been attached in supplementary materials. **Firstly, our massive Raman statistics were all conducted on**

suspended CNTs encapsulated within a stage. All spectra collections were preceded an Ar annealing treatment carried out at 450 °C, in case of a smear-out on the BWF characteristics caused by O₂ adsorption induced Fermi level shifts³⁷. These suspended CNTs after annealed treatment all best exhibit their intrinsic characteristics with a notably enhanced Raman intensity. Secondly, we treated the as-synthesized CNTs with cholate-D₂O solution (1% by weight) to further test the m-CNTs content. It has been verified that such electron-donating surfactants are effective in increasing the surface charge density of CNTs and strengthening the intensity of BWF regardless of whether they are resonant or not.²⁵ All these experiments have been introduced in detail in the Methods section with the specific data attached in the supplementary materials.

Comment 9

In general, Raman data is presented without specifying the wavelength used, and the effects of resonance on the resulting data are not discussed. There is clear potential for selection of metallic/semiconducting signals in this way, with some diameter dependence.

Reply: Thanks for your comments. As shown in Fig. 3b, outer diameters of DWNTs longer than 154 nm distribute in the range of 2.0~3.5 nm while inner ones range from 1.30 to 1.80 nm, which center around 1.7 nm. This corresponds to the E_{ii} ranges of 1.5-2.0 eV and 2.5-3.0 eV in the Kataura plot regarding the resonance of metallic components. However, the m-CNTs in the E_{ii} range of 2.5~3.0 eV are mostly families with high indexes ($2n+m=36, 39, 42$), which are seldom found in our as-grown CNTs. **Mono-wavelength lasers (2.33 eV, 1.96 eV, 1.58 eV) we used have covered most of the common tube families including the major distribution of m-CNTs. Therefore, it's relatively reliable to promise that there is at least one layer that resonates with the laser. Then both the profiles of two walls can be mostly available under Raman spectroscopy due to the quantum coupling.**¹⁴ But as for TWNTs, it becomes more complicated as the controversial interaction among three walls can be harder than quantum coupling. Resonance of a certain wall cannot permit to provide resonant information of all the walls. **Alternatively, we prefer to provide the Raman spectroscopy of the ~40 CNTs with specified chirality characterized by TEM, which can further testify the reliability of Raman for both DWNTs and TWNTs.** These spectra are also important references when we measured growth rates with the isotope labelling method.

According to the spectroscopic data collected from ~40 CNTs, there is correspondence in the quantity of m-CNTs between BWF shaped peaks and chirality identification under TEM. The BWF shaped G modes are sensitively available no matter which layers the metallic components lie in. Only all-semiconducting CNTs will give rise to Lorentzian G mode profiles featuring splitting peaks. Therefore, it proves that Raman spectroscopy is able to correctly identify all-semiconducting versus metallic-containing CNTs with high accuracy. We'd like to make the following revision (the words in green were copied from the main text while the red words are those after revising).

In the main text

...The aligned CNTs synthesized at other conditions (Supplementary Fig. 6) also demonstrated the ultrahigh s-purity at length >154 mm. It's worthy to note that there is no preference towards any kind of CNTs, single-walled (SWNTs), double-walled (DWNTs) or triple-walled (TWNTs) when discussing the rate selected growth. Despite of that the lasers we used can excite the ultralong CNTs within limited diameter distributions, the laser conditions, CNT structures, overlapping peaks, etc. will all influence the final observed numbers of components.⁸ It cannot permit to identify the wall numbers directly from the splitting G modes although the metallic components can be sensitively responded in the form of BWF peaks.

In the main text

The average TOF of s-CNTs ($\sim 1.5 \times 10^6 \text{ s}^{-1}$) is an order of magnitude higher than that of m-CNTs ($\sim 1.3 \times 10^5 \text{ s}^{-1}$) and also the highest among the reported industrial catalytic reactions. Besides, the resonance of these long tubes with the lasers was confirmed again from the correspondence of m-CNT (s-CNT) quantities between BWF (Lorentzian) shaped peaks and chirality identification under TEM (Supplementary Figs. 12-13).

In supporting materials

Supplementary Fig. 13 | Resonance of the MWNTs with laser irradiation wavelength.

a, Normalized Kataura plot: orange (purple) symbols, transition energies for metallic (semiconducting) SWNTs; Rectangular bars, laser excitation energies used in this work; Vertical lines: inner diameter range of DWNTs; Dotted line: predominated inner diameter. Outer diameters of DWNTs distribute in the range of 2.0~3.5 nm while inner ones range from 1.30 to 1.80 nm, which center around 1.7 nm. This corresponds to the E_{ii} ranges of 1.5~2.0 eV and 2.5~3.0 eV in the Kataura plot regarding the resonance of metallic components. However, the m-CNTs in the E_{ii} range of 2.5~3.0 eV are mostly families with high indexes ($2n+m=36, 39, 42$), which are seldom found in our as-grown CNTs. Monowavelength lasers (2.33 eV, 1.96 eV, 1.58 eV) have covered most of the common tube families including the major distribution of m-CNTs. Resonance of at least one layer with the laser determines the excitation of both the concentric layers due to quantum coupling.

Raman spectra of the m-¹²CNTs (b) and s-¹²CNTs (c) corresponding to those characterized under TEM.

Comment 10

It is suggested that the approach provides a 'scalable' approach to s-CNT production, but the absolute number of nanotubes remains low; there is no cheap bulk production here, though useful thin films may be produced. A comparison with other selective vapour growth work (eg from the Kauppinen group) is missing.

Reply: Thanks for your comments. In our point of view, it's a relative problem to discuss the absolute number of CNTs and the cost of production. **Previously, the areal density of aligned ultralong CNTs was generally lower than 1 CNT/ μm from a batch. Besides, there was almost no selectivity for m-CNTs or s-CNTs due to the indefinite interface between catalysts and CNTs.** However, in this work, we have largely increased the density by repeating the growth process with the help of preloaded catalysts. **The improved density in the best case can be even higher than 10 CNTs/ μm , which is a 10-time leap compared to the previous results.** At the same time, high concentration of s-CNTs have been discovered when the CNTs are long enough. We believe that these advancements have already achieved a great process compared to the previous results. Although more efforts are still necessary to prompt the scalable synthesis of ultralong CNTs, it's valuable to publish our periodical results in advance so as to receive more attention and support.

We don't consider it as a requisite to make comparison with other selective vapor growth methods. **Firstly, although we have improved the density of aligned ultralong CNTs, it's**

still far from the general level of thin film products. Kauppinen's group is famous for synthesizing CNT films with floating catalysts.³⁸ Their morphology, properties and functions are considerably different from our ultralong CNTs as synthesized. **Besides, it's not our focus to discuss the synthesis of high-density ultralong CNTs, as actually we have reported the preloading catalyst method to increase the density of aligned CNTs.** ³⁹ Here, we're just trying to highlight the novel phenomena and mechanism of the rate selected growth of s-CNTs. More contents about the scalable synthesis are on the going and we prefer to provide a better demonstration before they are deeply researched.

Comment 11

The implications of producing double or triple wall CNTs rather than single wall CNTs on device performance are not really considered. In general, these CNTs are less desirable, as they have different band gaps in each shell, likely generating a more complex response, likely less responsiveness to gating, and lower absolute band gaps.

Reply: Thanks for your questions. In fact, we hadn't intended to produce double or triple walled CNTs when synthesizing ultralong CNTs. During the past decades, we have focused more on the strategies to improve the length of CNTs without changing the chiral structures throughout. Because only on these perfect tubes will ballistic transport be easier to achieve, and devices integrated on these individual long tubes can possess the best uniformity of performances.⁴⁰ **During the process of research, we've discovered that few walled CNTs (mainly including SWNTs, DWNTs and TWNTs) are favored to reach macroscale length, especially the DWNTs and TWNTs.** Theoretically, these CNTs are not desirable to act as channels mainly due to their lower bandgaps of the outermost walls. **However, they surely have exhibited excellent electrical performances with high on/off ratio and current delivery, which has been demonstrated in other work as well.**^{16,41} We believe that there may be more complicated mechanism behind that is waiting for us to explore further. **After all, there lacks enough attention on the electrical performances of few walled CNTs because of their strongly coupled interaction among electron wave-functions.**

Comment 12

Generally, there is a substantial body of work here, which is no doubt of interest to those in the field. However, the relatively few new points of more general interest are not well-evidenced.

In places, the argument is hard to follow, due to non-idiomatic language, or non-sequiturs. For example, it is not clear in what sense there is a “seemingly huge Schrodinger’s cat state”. The title “self-purification” might be more clearly expressed, for example, as “Rate selected growth of pure semi-conducting CNT arrays”.

Reply: Thanks for your comments. In this work, our aim is just to highlight the phenomenon of bandgap dependent growth rate and the interlocking mechanism for growing perfect ultralong CNTs. **We believe that it’s an important fundamental study on the CNTs’ electrical properties using aligned ultralong CNTs with length up to decimeters and perfect structures**, which, we deeply believe that, will provide researchers with a deep and fundamental comprehension on **why longer tubes generally possess consistently perfect structures and high semiconducting selectivity** . Moreover, the data shown in the paper not only confirmed the expected high selectivity of perfect s-CNTs, but also provided a successful strategy to in-situ control the s-CNT purity, and also **revealed a new mechanism to achieve structural control despite of the vapor-liquid-solid growth mode** . We haven’t intended to expose too many ‘new’ points, which we think will lead to distraction of the focus from our core contents.

Schrodinger’s cat is a famous thought experiment in Physics devised by Erwin Schrodinger, which presents a hypothetical cat that may be simultaneously both alive and dead. This kind of state is also known as a quantum superposition, as a result of being linked to a random subatomic event that may or may not occur. ⁴² To mention it in our manuscript, we’re trying to take it as an analogy. Because, at the same time, it’s an equal probability event that if an individual CNT can be alive or dead after each dimer addition. **We assume that the CNT is such a macroscopic body composed of quantum coupled carbon atoms analogous to behaviors of Schrodinger’s cat. The whole life body of an individual CNT is compared to an iteration of multi-step uncertainties, which finally comprises a definite perfect structure with persistent bandgap.** By comparing with the Schrodinger’s cat, it will help to understand the complicated growth behaviors in the nano world, although there may be deeper physical mechanism behind that goes beyond what we’d expected.

We accept the advice as a better idea to make the title more clearly expressed. We have changed our title as ‘**Rate selected growth of the ultrapure semiconducting carbon nanotube arrays**’ (the words in green were copied from the main text while the red words are those after

revising).

In abstract

...Here, we demonstrate the **rate-selected** ultralong semiconducting CNT (s-CNT) arrays based on an interlocking between the atomic assembly rate and bandgap of CNTs. Rate analysis verified the Schulz-Flory (SF) distribution¹ for both metallic (m-) and s-CNTs, indicating their different decay rates as length increased. Quantitatively, a nearly ten-fold faster decay rate of m-CNTs, led to a spontaneous purification of 99.9999% s-CNTs at a length of 154 mm and the longest CNT can be 650 mm through an optimization of the reactor. Transistors fabricated on them delivered a high current of 14 $\mu\text{A}/\mu\text{m}$ with an on/off ratio around 10^8 and mobility over 4000 $\text{cm}^2/\text{V}\cdot\text{s}$. Our **rate-selected** strategy offers more freedom to *in-situ* control the CNT purity and provides a robust method to synthesize perfectly entangled condensate over a wide length scale.

In subtitle

Electrical properties of the **rate-selected ultralong s-CNT arrays.**

For further verification of the selectivity, we measured the electrical properties of the CNTs longer than 154 mm. ...

Reviewer #3

General comments

This manuscript reports observation of strong bandgap-dependent growth rate for few-wall carbon nanotubes, i.e., semiconducting tubes grow as much as 10 times faster than metallic tubes. Taking advantage of this observation, the authors prepared ultralong tubes (> 150 mm) that are predominantly semiconducting tubes and obtained impressive FET device performance. I recommend publication of the paper after the following concerns have been addressed in the revision.

Reply: Thank you for emphasizing the significance and giving highly positive comments on our work.

Comment 1

The formula describing SF distribution (eq. 1 and eq. 5) contain a quantity ($L-1$). What is the unit of this quantity? Is L a normalized length instead of the directly measured length in the unit of mm?

Reply: Thanks for proposing this problem. It's our carelessness to ignore this important detail. 'L' indicates the distance from the starting position of a substrate, so it should indicate a measured length, not a normalized one. We have chosen '1 mm' as a unit length in our previous work where the Schulz-Flory distribution was firstly demonstrated effective on ultralong CNTs.¹ Here, it's worthy of additional notes to make it much clearer. Please see the following revised text copied from the revised manuscript (the revised section are shown in red color).

In methods:

According to the SF distribution,

$$N_L = N_0 \alpha^{L-1} \quad (1)$$

Here, L denotes the distance from the starting position of a substrate, which is the measured length with a unit length of 1 mm. N_L indicates the total CNT quantity at position L .

Comment 2

In Figure 2i, data labelled as "sorted tubes" should be more properly labelled as "in situ synthesis

with chirality control" or something to that effect. Device performance of sorted tubes reported in ref 8 should be added to the figure for comparison.

Reply: Thanks for your kind reminder. We had planned to make comparisons between the performances of CNTs in situ synthesized and those solution processed. Now, we have revised the data labelling according to your advice and added ref 8 to the figure.

Reviewer #4

General comments

This was one of the more interesting papers that I have read in a while. The primary achievement is the demonstration that the growth rate of all-semiconducting few-walled carbon nanotubes is about 10 times faster than few-walled carbon nanotubes containing a metallic nanotube, during the floating synthesis of ultralong (100's of μm) nanotubes. The difference in growth rate insures that sufficiently downstream, predominantly semiconducting nanotubes are found. The authors demonstrate this phenomenon by characterizing the electronic-type of a statistically large body of nanotubes via Raman spectroscopy. TEM diffraction is also used to determine the (n,m) chirality of each wall in roughly 40 nanotubes to more definitely prove that the faster growing nanotubes are all-semiconducting. These are non-trivial syntheses and measurements. Electrical characterization data are also provided. If this synthesis strategy can be improved in the future to enable denser arrays of all-semiconducting nanotubes of smaller diameter (and larger bandgap) then it could enable practical implementations of semiconducting nanotubes in microelectronics. Based on the reported advances in synthesis and the depth of the characterization, this paper could eventually be publishable in Nature Communications. First, however, two major flaws must be corrected.

Reply: Thank you for your highly positive comments on our work. We have indeed done a lot of hard but careful work on synthesis and characterization of ultralong CNTs. It's our pleasure to receive your recognition on the significance and prospects of our work. We'd like to improve our work by following your professional advice.

Comment 1

The reporting of current/width and conductance/width for the interdigitated electrodes is nonsensical and must be removed throughout the abstract, text, and figures. The reason is that these parameters can be increased arbitrarily for any material (whether it be silicon, aligned carbon nanotubes, or random networks of solution processed nanotubes) by using interdigitated electrodes to multiple the effective channel width. The current/width and conductance/width will increase arbitrarily with the number of fingers in the electrodes. Therefore, why not create a million or billion or trillion fingers to skyrocket the claimed current/width or conductance/width towards infinity?

In high-performance integrated circuits, the current per width is an important parameter with respect to the capacitance per width. If the width is held constant then increases in current per width lead to higher performing (faster switching) transistors; thus the desire in industry to increase current per width. However, this assumes the capacitance per width is relatively invariant. In a transistor with an interdigitated set of electrodes, the capacitance will also increase with the number of fingers. Thus, there is no real gain to using interdigitated electrodes. The real channel width increases with the number of fingers (times 2). It is very deceiving to report current/width and conductance/width in interdigitated electrodes without correctly using the real channel width to normalize everything back to reality.

Please report only single-channel conductance (by normalizing by the actual interdigitated width) in: the abstract, manuscript text, Figures 2h,i, and Figures S8,9,10. The mobility and high on/off ratio are already impressive and demonstrate that the nanotubes are primarily semiconducting and of high crystalline quality.

Reply: Thanks for your kind remind. Actually, we didn't intend to provide deceiving recordings of the electrical performances with the integrated device structures. **We have made it rather outstanding as an important figure shown in Fig. 2d in order to be clearly understood.** Besides, the device shown in Fig. 2 is the only one based on the integrated structure while others in Fig. S8-10 are all based on normal device structures. In order to avoid misunderstanding, we have indicated in the manuscript that 'the device delivered a high current of 1.4 mA/ μm (corresponding to $\sim 14 \mu\text{A}/\mu\text{m}$ for single-channel transistor) with an on/off ratio of 10^8 '. By following your advice, we'd like to revise the statement in abstract as '**...Transistors fabricated on them delivered a high current of 14 $\mu\text{A}/\mu\text{m}$ with an on/off ratio around 10^8 and mobility over 4000 $\text{cm}^2/\text{V} \cdot \text{s}$** ' (the words in green were copied from the main text while the red words are those after revising).

Here, the reason why we highlight the integrated structures is that **this should be an important paradigm to utilize the ultralong CNTs, which has fully demonstrated their length strength capable of integrating multiple electrodes at the same time.** We have indicated it in the manuscript as '**...Compared with single-channel devices, this long-range interdigitated scheme fully utilizes the length strength of ultralong CNTs and compensates for the lower current output caused by lower density**'. Besides, we have demonstrated another

novel phenomenon that on/off ratio of the integrated aligned CNT device is nearly two times higher than that of individual CNTs. **We believe that such integrated device structures have taken important effects in improving both the on and off current. This is possibly helpful to exhibit the intrinsically high on/off ratio of perfect CNTs due to the lack of a smear-out by the minimum detection limit.** We have indicated it in the manuscript as 'Unlike the single-tube transistors, much higher on- and off-currents delivered by aligned CNTs can hardly be smeared out by the minimum ~0.1 pA-level noise detection limit. Thus, 10^8 may be the pristinely high on/off ratio of CNT transistors, which benefits from the improved density, cleanliness and alignment of these perfect ultralong s-CNTs'. Actually, there may exist deeper physical mechanism about the reason why pristinely high on/off ratio can be available with the integrated structure, but it surely has helped to discover higher values more close to reality. Therefore, we prefer to maintain some rather than revise them all.

Comment 2

The paper uses Raman spectroscopy as a means for gathering extensive statistics on metal versus semiconducting purity of the nanotubes. However, missing from the paper is a Kataura plot or a similar plot that shows that the laser energies used are reasonably close to the spectral resonances of the inner, middle, and outer walls of the range of nanotubes grown. If metallic nanotubes are present in a DWNT or TWNT but are not resonant with the laser then these metallic nanotubes may not show up in the Raman spectrum and the nanotubes may be falsely described as semiconducting types.

Alternatively, if the authors have the Raman spectra of all of the ~40 nanotubes that they characterized by TEM diffraction to determine the n,m of each wall, then it should be possible to prove (at least for these 40 nanotubes) that Raman spectroscopy is able to correctly identify all-semiconducting versus metallic-containing nanotubes, with high or 100% accuracy.

Reply: Thanks for your kind remind. In our work, we focused more on proving the sensitivity of BWF shaped G mode to metallic components, while this important problem of whether all the walls can resonate with the excitation energy was ignored. The advice you proposed here is really helpful to enhance the rigor of our work.

As shown in Fig. 3b, outer diameters of DWNTs longer than 154 nm distribute in the range of 2.0-3.5 nm while inner ones range from 1.30 to 1.80 nm, which center around 1.7 nm.

This corresponds to the E_{ii} ranges of 1.5~2.0 eV and 2.5~3.0 eV in the Kataura plot regarding the resonance of metallic components. However, the m-CNTs in the E_{ii} range of 2.5~3.0 eV are mostly families with high indexes ($2n+m=36, 39, 42$), which are seldom found in our as-grown CNTs. **Mono-wavelength lasers (2.33 eV, 1.96 eV, 1.58 eV) we used have covered most of the common tube families including the major distribution of m-CNTs. Therefore, it's relatively reliable to promise that there is at least one layer that resonates with the laser. Then both the profiles of two walls can be available under Raman spectroscopy due to the quantum coupling.**¹⁴ But as for TWNTs, it becomes more complicated as the controversial interaction among three walls can be harder than quantum coupling. Resonance of a certain wall cannot permit to provide resonant information of all the walls. **Alternatively, according to your advice, we prefer to provide the Raman spectroscopy of the ~40 CNTs with specified chirality characterized by TEM, which can further testify the reliability of Raman for both DWNTs and TWNTs.** These spectra are also important references when we measured growth rates with the isotope labelling method.

According to the spectroscopic data collected from ~40 CNTs, there is correspondence in the quantity of m-CNTs between BWF shaped peaks and chirality identification under TEM. The BWF shaped G modes are sensitively available no matter which layers the metallic components lie in. **Only all-semiconducting CNTs will give rise to Lorentzian G mode profiles featuring splitting peaks.** Therefore, it proves that Raman spectroscopy is able to correctly identify all-semiconducting versus metallic-containing CNTs with high accuracy. By taking your advice, we'd like to make the following revision (the words in green were copied from the main text while the red words are those after revising).

In the main text

...The aligned CNTs synthesized at other conditions (Supplementary Fig. 6) also demonstrated the ultrahigh s-purity at length >154 μ m. It's worthy to note that there is no preference towards any kind of CNTs, single-walled (SWNTs), double-walled (DWNTs) or triple-walled (TWNTs) when discussing the rate selected growth. **Despite of that the lasers we used can excite the ultralong CNTs within limited diameter distributions, the laser conditions, CNT structures, overlapping peaks, etc. will all influence the final observed numbers of components.**⁸ It cannot permit to identify the wall numbers directly from the splitting G modes although the metallic components can be sensitively

responded in the form of BWF peaks.

In the main text

The average TOF of s-CNTs ($\sim 1.5 \times 10^6 \text{ s}^{-1}$) is an order of magnitude higher than that of m-CNTs ($\sim 1.3 \times 10^5 \text{ s}^{-1}$) and also the highest among the reported industrial catalytic reactions. Besides, the resonance of these long tubes with the lasers was confirmed again from the correspondence of m-CNT (s-CNT) quantities between BWF (Lorentzian) shaped peaks and chirality identification under TEM (Supplementary Figs. 12-13).

In supporting materials

Supplementary Fig. 13 | Resonance of the MWNTs with laser irradiation wavelength.

a, Normalized Kataura plot: orange (purple) symbols, transition energies for metallic (semiconducting) SWNTs; Rectangular bars, laser excitation energies used in this work; Vertical lines: inner diameter range of DWNTs; Dotted line: predominated inner diameter. Outer diameters of DWNTs distribute in the range of 2.0~3.5 nm while inner ones range from 1.30 to 1.80 nm, which center around 1.7 nm. This corresponds to the E_{ii} ranges of 1.5~2.0 eV and 2.5~3.0 eV in the Kataura plot regarding the resonance of metallic components. However, the m-CNTs in the E_{ii} range of 2.5~3.0 eV are mostly families with high indexes ($2n+m=36, 39, 42$), which are seldom found in our as-grown CNTs. Mono-

wavelength lasers (2.33 eV, 1.96 eV, 1.58 eV) have covered most of the common tube families including the major distribution of m-CNTs. Resonance of at least one layer with the laser determines the excitation of both the concentric layers due to quantum coupling.

Raman spectra of the m-¹²CNTs (b) and s-¹²CNTs (c) corresponding to those characterized under TEM.

Comment 3

Please report Drain current versus Gate Voltage characteristics for higher V_D at least up to 1V. Prior work (Zhu et al. Sci. Adv. 2016 2 e1601572) has shown that high on/off ratio can only be obtained at small V_D . If high on/off ratio cannot be obtained at higher V_D for example of 0.5V then the reader should be made aware of this deficiency which may originate due to the small bandgap of the outer nanotubes.

Reply: Thanks for your kind remind. We admit that your advice is really helpful to improve our work and explore deeper about the mechanism for high on/off ratio of ultralong CNTs. **Actually, we have demonstrated the transfer characteristics of individual ultralong CNTs at $V_D=0.5$ V in our prior work (fig. S10 in Zhu et al. Sci. Adv. 2016 2 e1601572).⁴¹ The on/off ratio is generally around 10^5 while it can be higher as 10^6 - 10^7 when the devices are operated at $V_D=0.1$ V, which has been reported in this work shown in Figs. S9-10. Just as you said, the on/off ratio will generally decrease with the increment of V_D . However, it would be much more difficult to provide the transfer characteristics at higher V_D of the device shown in Fig. 2h. Due to the integrated device structure and the long contact length, the current delivery of densely aligned CNTs will be higher than the limit that the detector can withstand.**

Therefore, these data are surely hard to obtain although we had tried. But we believe that the data for individual ultralong CNTs we have reported so far both in this work and the prior work can give enough insights on the electrical characteristics and best demonstrate your point.

Other important points:

Comment 4

The paper states “The total single-tube capacitance can be viewed as the harmonic mean of the electrostatic gate capacitance and the quantum capacitance, described by the equation $C_{g-1} = C_{g,el-1} + C_{q-1}$. The electrostatic gate capacitance is $C_{g,el} = 7 \times 10^{-9}$ F/m, significantly larger than the quantum capacitance $C_q \approx 4 \times 10^{-10}$ F/m and the $C_{g,el}$ term can be ignored.”

However, I calculate that $C_{g,el} = 3 \times 10^{-11}$ F/m (wire-plate separated by 800 nm, dielectric constant 3.9, wire radius = 1.25 nm). Certainly, the C_q should not dominate until the dielectric constant becomes very (< 10 nm) thin.

Reply: Thanks for your efforts on revising our calculations on the electrostatic gate capacitance. Certainly, we made a mistake when we presented the calculation results of $C_{g,el}$. In fact, we calculated $C_{g,el}$ according to the formula $C_{g,el} = 2\pi\epsilon\epsilon_0/\ln(1 + \frac{2D}{R})$, which was reported in the reference (doi:10.1021/nl025639a).⁴³ By substituting $\epsilon=3.9$, $D=800$ nm, $R=2.4$ nm (estimated based on the statistics on ~40 CNTs longer than 154 nm), we obtained $C_{g,el}=3.1 \times 10^{-11}$ F/m. Then the electrostatic gate capacitance will be $C_g \approx C_{g,el}=3.1 \times 10^{-11}$ F/m with the ignorance of quantum capacitance $C_q \approx 4 \times 10^{-10}$ F/m. Having calculated further, we can get the mobility of the transfer characteristics shown in Fig. S10a concentrates around 4451 cm²/Vs. **Therefore, there is no problem in the calculations of CNT mobility but it was indeed incorrectly expressed in the methods. Maybe we were confusing our result with the one reported in that reference when we wrote the manuscript. Because the electrostatic gate capacitance reported in that reference was 7×10^{-9} F/m.** For whatever the reason, thanks for pointing out our mistake and we'd like to revise the manuscript based on your advice.

In methods

...Field effect mobility (μ_e) of an individual CNT can be calculated according to the

equation $\mu = \frac{L_{ch}}{C} \frac{dG_{on}}{dV_{gs}}$, where C is the single-tube capacitance. The total single-tube capacitance can be viewed as the harmonic mean of the electrostatic gate capacitance and the quantum capacitance, described by the equation $C_g^{-1} = C_{g,el}^{-1} + C_q^{-1}$. The electrostatic gate capacitance is $C_{g,el} \approx 3.1 \times 10^{-11}$ F/m, significantly smaller than the quantum capacitance $C_q \approx 4 \times 10^{-10}$ F/m and the C_q term can be ignored. Then the single-tube mobility can be obtained based on the transfer characteristics of the transistor and its device structural parameters.

Comment 5

Please comment in the text how it is possible to realize such large on current - off current switching (>1E6 or as high as 1E8) even though energy gap of outerwall NT is so small (~0.2 eV). The on current will be proportional to the VG times the capacitance = 30V * 3E-11 F/m = 6E7 holes/cm. The off current will be proportional to the thermally generated carrier density. According to Akinwande et al. IEEE TRANSACTIONS ON ELECTRON DEVICES, VOL. 55, NO. 1, JANUARY 2008, this should be about 1E4 holes/cm for a nanotube with 0.26 eV energy gap at 300K (see Figure 4). The ratio between the on current and off current should be approximately 6E3 for a 0.26eV nanotube and less for nanotubes with larger outerwalls that have energy gaps less than 0.26eV.

Reply: Thanks for your careful verification on the extraordinary on/off ratio. Actually, we have been confused about this issue for a long time as well. Because this is not the first time that we discovered the high on/off ratio of few walled CNTs. **Previously in our prior work, we also reported close values $\sim 10^5$ - 10^6 for devices of individual ultralong CNTs with different experimental setup.**⁴¹ In this work, higher on/off ratio has been discovered once again for devices based on aligned few walled CNTs. These results surely deviate from the earlier theoretical calculations reported by Akinwande et al.. But it remains controversial and open why the few walled CNTs can achieve such high on/off ratio. **Still, there have been fewer studies on the electrical properties of large-diameter or few walled CNTs due to the complicated multi-body physics of MWNTs.** Although we're unable to provide strict theories to interpret the low off-current, we'd like to offer some clues about the deviation from theoretical calculations. **Firstly, what Akinwande care about is the intrinsic carrier density of achiral SWNTs, while we pay more attention to the chiral few walled CNTs.** These

incommensurate structures among walls bring strongly correlated coupling, which will significantly influence the electronic behaviors at the off state.^{14,44} Besides, it has been demonstrated that direct molecular dynamics or Monte Carlo simulations fail to produce non-defective CNT structures of a well-defined diameter and recognizable chirality.²⁷ **It seems that a physical theory bridging the gap between atomistic dynamics and macroscopic scales is needed to interpret both the experiments and simulations. We assume that the energy relationship isn't enough for Dirac-type CNTs, additional dimensionality of momentum should also be considered.** So far, there has been limited understanding on the few walled CNTs from a physical point of view. Further study is needed to clarify this issue. We're expecting more collaborations from other research groups skilled at electronics related theories after publishing this work.

Comment 6

Fig. 2h. The figure indicates V_{DS} is 0.1V. The caption indicates 1V.

Reply: We are so sorry to have made this mistake. The actual V_{DS} should be 0.1 V and we have revised it in the caption of Fig. 2h.

Comment 7

Fig. 2e. The scale bar reads 10 mm when probably this should be μm . Same with figure g.

Reply: Thanks for your careful examination and kind reminder. It's our carelessness to make this mistake. We have revised the units above the scale bar in Fig. 2e as 10 μm .

Comment 8

Fig. S10e. Units are A but probably they should be μA .

Reply: Thanks for your kind reminder. We have revised the units in Fig. S10e as μA .

[Reference]

- 1 Zhang, R. et al. Growth of Half-Meter Long Carbon Nanotubes Based on Schulz–Flory Distribution. *ACS Nano* **7**, 6156-6161 (2013).
- 2 Gao, M. et al. Structure determination of individual single-wall carbon nanotubes by nanoarea electron diffraction. *Appl. Phys. Lett.* **82**, 2703-2705 (2003).
- 3 Lu, W. et al. A Scanning Probe Microscopy Based Assay for Single-Walled Carbon Nanotube Metallicity. *Nano Lett.* **9**, 1668-1672 (2009).
- 4 Aravind, V. et al. Toward single-chirality carbon nanotube device arrays. *ACS Nano* **4**, 2748 (2010).
- 5 Wilder, J. W. G., Venema, L. C., Rinzler, A. G., Smalley, R. E. & Dekker, C. Electronic structure of atomically resolved carbon nanotubes. *Nature* **391**, 59-62 (1998).
- 6 Dresselhaus, M. S., Dresselhaus, G., Saito, R. & Jorio, A. Raman spectroscopy of carbon nanotubes. *Phys. Rep.* **409**, 47-99 (2005).
- 7 Villalpando-Paez, F. et al. Raman Spectroscopy Study of Isolated Double-Walled Carbon Nanotubes with Different Metallic and Semiconducting Configurations. *Nano Lett.* **8**, 3879-3886 (2008).
- 8 Levshov, D. I. et al. Accurate determination of the chiral indices of individual carbon nanotubes by combining electron diffraction and Resonant Raman spectroscopy. *Carbon* **114**, 141-159 (2017).
- 9 Paillet, M. et al. Probing the structure of single-walled carbon nanotubes by resonant Raman scattering. *Phys. Status Solidi* **247**, 2762-2767 (2010).
- 10 Tian, Y., Jiang, H., Laiho, P. & Kauppinen, E. I. Validity of Measuring Metallic and Semiconducting Single-Walled Carbon Nanotube Fractions by Quantitative Raman Spectroscopy. *Anal. Chem.* **90**, 2517-2525 (2018).
- 11 Naumov, A. V., Ghosh, S., Tsyboulski, D. A., Bachilo, S. M. & Weisman, R. B. Analyzing Absorption Backgrounds in Single-Walled Carbon Nanotube Spectra. *ACS Nano* **5**, 1639-1648 (2011).
- 12 Tulevski, G. S., Franklin, A. D. & Afzali, A. High Purity Isolation and Quantification of Semiconducting Carbon Nanotubes via Column Chromatography. *ACS Nano* **7**, 2971-2976 (2013).

- 13 Ferrari, A. C. *et al.* Raman Spectrum of Graphene and Graphene Layers. *Phys. Rev. Lett.* **97**, 187401 (2006).
- 14 Liu, K. *et al.* Quantum-coupled radial-breathing oscillations in double-walled carbon nanotubes. *Nat. Commun.* **4**, 1375 (2013).
- 15 Hirschmann, T. C. *et al.* Role of Intertube Interactions in Double- and Triple-Walled Carbon Nanotubes. *ACS Nano* **8**, 1330-1341 (2014).
- 16 Wen, Q. *et al.* 100 mm Long, Semiconducting Triple-Walled Carbon Nanotubes. *Adv. Mat.* **22**, 1867-1871 (2010).
- 17 Hashimoto, A. *et al.* Atomic Correlation Between Adjacent Graphene Layers in Double-Wall Carbon Nanotubes. *Phys. Rev. Lett.* **94**, 045504 (2005).
- 18 Levshov, D. *et al.* Experimental Evidence of a Mechanical Coupling between Layers in an Individual Double-Walled Carbon Nanotube. *Nano Lett.* **11**, 4800-4804 (2011).
- 19 Ghedjatti, A. *et al.* Structural Properties of Double-Walled Carbon Nanotubes Driven by Mechanical Interlayer Coupling. *ACS Nano* **11**, 4840-4847 (2017).
- 20 Zhang, R. *et al.* Superlubricity in centimetres-long double-walled carbon nanotubes under ambient conditions. *Nat. Nanotechnol.* **8**, 912-916 (2013).
- 21 Hirahara, K. *et al.* Chirality correlation in double-wall carbon nanotubes as studied by electron diffraction. *Phys. Rev. B* **73**, 195420 (2006).
- 22 Gao, M., Zuo, J. M., Zhang, R. & Nagahara, L. A. J. J. o. M. S. Structure determinations of double-wall carbon nanotubes grown by catalytic chemical vapor deposition. *J. Mat. Sci.* **41**, 4382-4388 (2006).
- 23 Ding, F. *et al.* The Importance of Strong Carbon-Metal Adhesion for Catalytic Nucleation of Single-Walled Carbon Nanotubes. *Nano Lett.* **8**, 463-468 (2008).
- 24 Wang, X. *et al.* Fabrication of Ultralong and Electrically Uniform Single-Walled Carbon Nanotubes on Clean Substrates. *Nano Lett.* **9**, 3137-3141 (2009).
- 25 Blackburn, J. L., Engtrakul, C., McDonald, T. J., Dillon, A. C. & Heben, M. J. Effects of Surfactant and Boron Doping on the BWF Feature in the Raman Spectrum of Single-Wall Carbon Nanotube Aqueous Dispersions. *J. Phys. Chem. B* **110**, 25551-25558 (2006).
- 26 Ding, F., Harutyunyan, A. R. & Yakobson, B. I. Dislocation theory of chirality-controlled nanotube growth. *P. Natl. Acad. Sci. USA* **106**, 2506 (2009).

- 27 Artyukhov, V. I., Penev, E. S. & Yakobson, B. I. Why nanotubes grow chiral. *Nat. Commun.* **5**, 4892 (2014).
- 28 Yang, F. et al. Chirality-specific growth of single-walled carbon nanotubes on solid alloy catalysts. *Nature* **510**, 522-524 (2014).
- 29 Zhang, S. et al. Arrays of horizontal carbon nanotubes of controlled chirality grown using designed catalysts. *Nature* **543**, 234-238 (2017).
- 30 Deshpande, V. V. et al. Mott Insulating State in Ultraclean Carbon Nanotubes. *Science* **323**, 106 (2009).
- 31 Huang, S., Woodson, M., Smalley, R. & Liu, J. Growth Mechanism of Oriented Long Single Walled Carbon Nanotubes Using "Fast-Heating" Chemical Vapor Deposition Process. *Nano Lett.* **4**, 1025-1028 (2004).
- 32 Cao, Q. et al. Gate capacitance coupling of singled-walled carbon nanotube thin-film transistors. *Appl. Phys. Lett.* **90**, 023516 (2007).
- 33 Sun, D.-m. et al. Flexible high-performance carbon nanotube integrated circuits. *Nat. Nanotechnol.* **6**, 156-161 (2011).
- 34 Franklin, A. D. The road to carbon nanotube transistors. *Nature* **498**, 443 (2013).
- 35 Qin, X. et al. Growth of Semiconducting Single-Walled Carbon Nanotubes by Using Ceria as Catalyst Supports. *Nano Lett.* **14**, 512-517 (2014).
- 36 Li, W.-S. et al. High-Quality, Highly Concentrated Semiconducting Single-Wall Carbon Nanotubes for Use in Field Effect Transistors and Biosensors. *ACS Nano* **7**, 6831-6839 (2013).
- 37 Nguyen, K. T., Gaur, A. & Shim, M. Fano Lineshape and Phonon Softening in Single Isolated Metallic Carbon Nanotubes. *Phys. Rev. Lett.* **98**, 145504 (2007).
- 38 Liao, Y. et al. Direct Synthesis of Colorful Single-Walled Carbon Nanotube Thin Films. *J. Am. Chem. Soc.* **140**, 9797-9800 (2018).
- 39 Xie, H. et al. Preloading catalysts in the reactor for repeated growth of horizontally aligned carbon nanotube arrays. *Carbon* **98**, 157-161 (2016).
- 40 Chen, Z. et al. An integrated logic circuit assembled on a single carbon nanotube. *Science* **311**, 1735-1735 (2006).
- 41 Zhu, Z. et al. Acoustic-assisted assembly of an individual monochromatic ultralong carbon nanotube for high on-current transistors. *Sci. Adv.* **2** (2016).

- 42 Schrödinger, E. J. N. Die gegenwärtige Situation in der Quantenmechanik. *Naturwissenschaften* **23**, 823-828 (1935).
- 43 Rosenblatt, S. et al. High Performance Electrolyte Gated Carbon Nanotube Transistors. *Nano Lett.* **2**, 869-872 (2002).
- 44 Liu, K. et al. Van der Waals-coupled electronic states in incommensurate double-walled carbon nanotubes. *Nat. Phys.* **10**, 737-742 (2014).

Reviewers' Comments:

Reviewer #1:

Remarks to the Author:

I am content with the response from the authors and revised manuscript. Since the authors well address all of my questions and comments, I think this manuscript can be published in Nature Communications.

Reviewer #2:

Remarks to the Author:

The authors have made some modest corrections to the manuscript addressing some minor errors, and usefully adding further information about Raman resonance. Whilst acknowledging the issues raised in the response, they have not fully clarified the manuscript to avoid misleading the reader. Specifically, the following points have NOT been adequately addressed, and can easily be included:

- The abstract still claims the work "led to a spontaneous purification of 99.9999% s-CNTs at a length of 154 nm". The authors MUST make clear that this value is a prediction or extrapolation of the growth model, not a proven value. In the response, the authors say "We hadn't intended to provide the specific purity for s-CNTs", but the text, as written, continues to make this claim.
- Similarly, the authors should not state that "this on/off ratio is ~100 times higher than that of single-tube transistors". It is only possible to say that "this on/off ratio is ~100 times higher than that which can be MEASURED for single-tube transistors." The work therefore extends the upper bound to the maximum on-off ratio that can be obtained for CNTs. In fact, this high value is difficult to understand, as flagged by reviewer 4, but some of this discussion would be useful to include in the paper.
- The authors have retained the misleading interdigitated current density in figure 2i. As explained by reviewer 4, this value is arbitrary and should be removed from a comparison of channel properties. A comment in the text about the absolute value obtained, and how it helps with measurements, would be appropriate.
- Reviewer 4's helpful comments about capacitance should be included within the discussion to explain that the higher current available from the interdigitated electrodes is useful for exploring limits of CNT performance but not for practical devices.
- A further discussion about the data summarised in 2a should consider the question of the drain voltages used. The authors should include a frank explanation of the 0.1V drain voltage used, the relationship to the band gap of these CNTs, and the associated limitations. The general reader will be interested in the practical significance of the findings, even if the authors are primarily focussed on the rate question.
- The authors now state "It's worthy to note that there is no preference towards any kind of CNTs, single-walled (SWNTs), double-walled (DWNTs) or triple-walled (TWNTs) when discussing the rate selected growth." But what they mean is (I think) that their experiment is not sensitive to any preference. In fact, there is likely a significant difference, given the disappearance of the SWNTs.

More minor points:

It would be clearer if at the point in the manuscript claiming 99.99% s-CNT purity ("A G mode collective (Supplementary Fig. 4) statistically counted from ~104 tubes longer than 154 nm, displayed neither BWF nor D band signals") mentioned explicitly that the careful Raman analysis is a summation of spectra from individual CNTs, where each spectrum is proven to be clearly sensitive to electron character, using the ED data. It's clear in the SI, but not as worded in the main text.

The claim that the TOF is "also the highest among the reported industrial catalytic

reactions." needs some reference(s); is the effect just the higher temperature of the process, or something fundamental?

The Schrodinger's cat analogy does not make sense. The grown part of the tube exists unchanged whether or not the end is still growing. The tail of the cat is different depending on the quantum event, the tail of the CNT is the same. The analogy serves no purpose.

The language is difficult to read throughout and would benefit from a thorough review.

Reviewer #4:

Remarks to the Author:

Rereview:

The paper has been improved. The updates have partially addressed the revisions that I previously requested. However, there are still deficiencies that remain to be addressed. Overall, my opinion is that this is a very interesting paper worth publishing in Nature Communications, provided these deficiencies can be addressed.

Follow-up on previous comment 1)

The misleading reporting of current per width for the interdigitated arrays has been eliminated in the abstract. However, it still appears in the y-axis of Fig. 2h, in Fig. 2f as the solid red stars, and in the text. Please update as follows:

a) Update the data shown in Fig. 2h y-axis to report current per width where the width is the physical width of each channel times the number of interdigitated channels.

b) Remove the solid red stars in Fig. 2f.

c) Fix the sentence "At a drain-source bias (V_{DS}) of 0.1 V, the device delivered a high current of 1.4 mA/ μm ", which does not make sense because the channel width is actually much more. The authors follow up this sentence with "(corresponding to $\sim 14 \mu\text{A}/\mu\text{m}$ for single-channel..." It is this 14 $\mu\text{A}/\mu\text{m}$ number that should be reported in the first place.

The authors' argument that "...Compared with single-channel devices, this long-range interdigitated scheme fully utilizes the length strength of ultralong CNTs and compensates for the lower current output caused by lower density" is misleading. The increase in current with more interdigitations is NOT a property unique to their ultralong CNTs. The current of any material: silicon, oxides, graphene, 2D materials, random networks of CNTs, aligned arrays of short CNTs – all will increase using the interdigitation scheme linearly with the number of electrodes. There is no limit to this effect – the more interdigitations the more the current. This "strategy" of using interdigitations to artificially increase current per width has no merit and would only harm the field and scientific if established as a means for making device performance *seem* better.

The ability to determine a higher on/off ratio is still true if both on and off current are properly normalized by the actual channel width (width times the number of channels in parallel)... so this is not justification for using the falsely normalized current per width.

Follow up on previous comment 5)

Without a good justification of how the on/off ratio is so high, the authors should at minimum provide more information about their measurement setup and its abilities to ensure the reader and community that their measurements of off-current are not being skewed by unexpected factors. Please (a) provide measures of the gate current I_g as a function of V_g in devices in which I_{ds} is also measured as a function of V_g . Depending on measurement setup, it is possible to be sensitive to I_{ds} minus I_g when expecting to be sensitive only to I_{ds} , which can result in artificially low measures of I_{off} .

Miscalibration (Which can be tested in FET devices without nanotubes) should also be characterized.

New comment 9) Some of the new sentences need more editing to improve readability and grammar including.

"Besides, the resonance of these long tubes with the lasers was confirmed again from the correspondence of m-CNT (s-CNT) quantities between BWF (Lorentzian) shaped peaks and chirality identification under TEM (Supplementary Figs. 12-13)."

"Despite of that the lasers we used can excite the ultralong CNTs within limited diameter distributions, the laser conditions, CNT structures, overlapping peaks, etc. will all influence the final observed numbers of components¹⁹."

New comment 10) Possible typos in:

Fig. S10d caption "d, The histogram showing the distribution of on-state conductance as measured in c. e, Collection of output characteristics with an applied VGS of 15 V". 15 V should be -15 V, presumably.

"b, Output characteristic of the transistor measured with ascending VGS from 11 V to 17 V at a step of 1 V." should be -11 to -17 at a step of -1 V, presumably.

Response to Reviewers' Comments

Reviewer #2

General comments

The authors have made some modest corrections to the manuscript addressing some minor errors, and usefully adding further information about Raman resonance. Whilst acknowledging the issues raised in the response, they have not fully clarified the manuscript to avoid misleading the reader. Specifically, the following points have NOT been adequately addressed, and can easily be included:

Reply: Thanks for your professional evaluation and warmhearted feedback. Last time after we received your comments, we had a heated discussion about what you had addressed, which we admired so much and surely provided us with considerable constructive directions. Also, those opinions could remarkably help enlighten our work in the future. We spared no efforts in exchanging our ideas and revising the report in our last response expecting to receive your approval. Although it's a pity that our effort is not enough to get this report qualified and satisfied, we're so pleased to enjoy another chance to discuss further the academics of our work. And it's our pleasure to improve the work by taking your professional advice. We're sincerely expecting to get this report qualified after this round of revision as this might be the final chance for us.

Comment 1

The abstract still claims the work "led to a spontaneous purification of 99.9999% s-CNTs at a length of 154 nm". The authors MUST make clear that this value is a prediction or extrapolation of the growth model, not a proven value. In the response, the authors say "We hadn't intended to provide the specific purity for s-CNTs", but the text, as written, continues to make this claim.

Reply: We feel sorry that our statement is not inspiring and even brings more misunderstanding. With the same point of view, we strongly agree with you that it's important to respect the facts. **It's necessary to claim the predictive characteristics of our reported value so that it won't result in any false information or even mislead our general readers.** We are deeply impressed by your rigorous attitude towards academics and you have inadvertently set a

respectable example to us all. We hope that the following revision can be appropriate (the words in green were copied from the main text while the red words are those after revising).

.....Quantitatively, a nearly ten-fold faster decay rate of m-CNTs, led to a spontaneous purification of the predicted 99.9999% s-CNTs at a length of 154 mm and the longest CNT can be 650 mm through an optimization of the reactor.

Comment 2

Similarly, the authors should not state that “this on/off ratio is ~100 times higher than that of single-tube transistors”. It is only possible to say that “this on/off ratio is ~100 times higher than that which can be MEASURED for single-tube transistors.” The work therefore extends the upper bound to the maximum on-off ratio that can be obtained for CNTs. In fact, this high value is difficult to understand, as flagged by reviewer 4, but some of this discussion would be useful to include in the paper.

Reply: Thanks for your nice and professional comments. This statement definitely tends to cause misunderstanding and it seems controversial to compare between the on/off ratio of aligned and individual CNTs. Therefore, **we’d like to remove such statements in order to avoid any controversy.** Besides, we believe that **it’s more significant to directly report the measurement results of the devices fabricated on individual ultralong CNTs. Because they have only been displayed in the Supplementary Materials but not in the main text, which will be neglected but still are very important.** Therefore, we’d like to supplement them in the main text as additional evidence to support the high semiconducting purity and perfect structure of these CNTs (the red words are those after revising while the red words with the bold and underlining font are revisions aimed to this part).

...Therefore, we normalized the performances of the ultralong-CNT-array transistors for the whole length of the channel and compared them with those previously reported¹⁻⁷ in Fig. 2i. It indicates a significant improvement of the s-CNT purity. The superior performance of the device based on aligned CNTs comes from the remarkable electrical properties of individual ultralong CNTs. We measured 452 transistors each built on a single CNT with the length over 154 mm (Supplementary Figs. 11-12). On average, they have exhibited excellent electrical performances, such as the on/off ratio around 10^6 , on-state conductance of 16 μS and mobility of $4451 \text{ cm}^2/\text{V}\cdot\text{s}$.

Besides, we agree with you to **add the discussions flagged by reviewer 4 in the last review into the main text** (the red words are those after revising while the red words with the bold and underlining font are revisions aimed to this part).

...The superior performance of the device based on aligned CNTs comes from the remarkable electrical properties of individual ultralong CNTs. We measured 452 transistors each built on a single CNT with the length over 154 nm (Supplementary Figs. 11-12). On average, they have exhibited excellent electrical performances, such as the on/off ratio around 10^6 , on-state conductance of 16 μS and mobility of 4451 $\text{cm}^2/\text{V}\cdot\text{s}$. However, it's still hard to understand such high on/off ratio for transistors fabricated on the few-walled CNTs with smaller bandgap (~0.2 eV). The on current, $\sim 6 \times 10^7$ holes/cm, is estimated to be the product of V_{GS} and capacitance while the off current is proportional to the thermally generated carrier density⁸. Theoretically, the on/off ratio should be lower than 6×10^3 for a CNT with the outermost bandgap less than 0.26 eV, which is 10^5 times lower than that we actually measured. Direct molecular dynamics or Monte Carlo simulations also failed to analyze this anomaly due to the difficulties in simulating non-defective CNT structures of a well-defined diameter and recognizable chirality^{9,10}. It seems that a physical theory bridging the gap between atomistic dynamics and macroscopic scales is needed to interpret both the experiments and simulations. We assume that the energy relationship is not enough for Dirac carbon species¹¹, and additional dimensionality of momentum might be another important factor we had not expected.

Comment 3

The authors have retained the misleading interdigitated current density in figure 2i. As explained by reviewer 4, this value is arbitrary and should be removed from a comparison of channel properties. A comment in the text about the absolute value obtained, and how it helps with measurements, would be appropriate.

Reply: Thanks for your comments. We feel so sorry that our way of displaying the data has resulted in so much controversy and misunderstanding in the review process. We promise that it's not our intention to make the data seem better but we're seeking the best way to demonstrate the length strength of ultralong CNTs in the electrical applications. We admire the penetrating comments of you and reviewer 4 so much, making us recognize the seriousness

and importance to report the absolute data. Therefore, we're wondering if it's appropriate to revise our manuscript as follows. **We have put the single-channel current in the first place before reporting the absolute result for the whole device.** Besides, **we have added additional interpretation on the reason why we used the interdigitated scheme.** Because such large electrode area can still retain a high on/off ratio and mobility, we consider it as another important evidence demonstrating the stable semiconducting property and perfect structure of the CNTs in a long length scale. At the same time, **we have updated the data shown in Fig. 2h y-axis to report current per width where the width is the physical width of each channel times the number of interdigitated channels.** The data symbolled by solid red stars have been removed and we labeled that area as an outstanding 'superb' , which indicates an expected target we hope to achieve in the near future. Besides, **we have added more information about our measurement setup, which helps to demonstrate how it helps to increase the resolution and fulfill our measurement** (the words in green were copied from the main text while the red words are those after revising).

In the main text

...Prototypical ~100 parallel transistors (**Figs. 2d-g**) were fabricated on aligned long tubes contacted with interdigitated electrodes. At a drain-source bias (V_{DS}) of -0.1 V, the device delivered a high width-normalized current of ~14 $\mu\text{A}/\mu\text{m}$ for a single channel (corresponding to ~42 mA output for the whole device) with an on/off ratio of 10^8 and field effect mobility¹² >4000 $\text{cm}^2/\text{V}\cdot\text{s}$ (**Fig. 2h**)....

Covering 1.2-mm-long CNTs, this interdigitated device can still retain a high on/off ratio and mobility, which effectively demonstrates the stable semiconducting property and perfect structure of the long CNTs.

In Methods

Fabrication and measurement of the transistors. Ultralong CNTs were grown on a heavily n-doped silicon wafer with an 800-nm SiO_2 top layer..... Electronic measurements were carried out by applying drain and gate voltages relative to the source electrode with a Keithley 4200A parameter analyzer at room temperature in air. The drain current was measured with a Keithley 4200-PA amplifier. The low current measurement capabilities of any SMU can be extended by adding an optional 4200-PA preamplifier. The preamplifier provides 10 nA resolution by

effectively adding five current ranges to either SMU model.

Comment 4

Reviewer 4's helpful comments about capacitance should be included within the discussion to explain that the higher current available from the interdigitated electrodes is useful for exploring limits of CNT performance but not for practical devices.

Reply: This point that you have addressed really impresses us and thanks for reminding us of the possible effects of capacitance on the limits of CNT performance. Just as what reviewer 4 said in the last review, 'In a transistor with an interdigitated set of electrodes, the capacitance will also increase with the number of fingers. Thus, there is no real gain to using interdigitated electrodes.' Due to the synchronous increase of capacitance with the interdigitated scheme, it will possibly cause errors when discussing the current delivery of the whole device. Therefore, according to reviewer 4's suggestions in this review, **we have modified the data in Fig. 2h only reporting the performance for a single channel after necessary normalization.** But what you have proposed here indicates the potential reason for the excellent performance of interdigitated devices in contrast to single-CNT devices. **We agree with you to supplement some additional information about the capacitance** (the red words are those after revising while the red words with the bold and underlining font are revisions aimed to this part).

In the main text

...Covering 1.2-mm-long CNTs, this interdigitated device can still retain a high on/off ratio and mobility, which effectively demonstrates the stable semiconducting property and perfect structure of the long CNTs. However, in a transistor with an interdigitated set of electrodes, the capacitance will increase with the number of fingers so that there is no real delivery gain regarding the use of interdigitated electrodes. The higher current available from the interdigitated electrodes is useful for exploring the limits of CNT performance but not for the practical devices. Therefore, we normalized the performances of the ultralong-CNT-array transistors for the whole length of the channel and compared them with those previously reported¹⁻⁷ in Fig. 2i. It indicates a significant improvement of the s-CNT purity.

Comment 5

A further discussion about the data summarised in 2a should consider the question of the drain voltages used. The authors should include a frank explanation of the 0.1V drain voltage used,

the relationship to the band gap of these CNTs, and the associated limitations. The general reader will be interested in the practical significance of the findings, even if the authors are primarily focussed on the rate question.

Reply: Thanks for pointing out this important problem. We understand the point that you feel confused or worried about may be related to the particularity of this drain voltage. **Actually, there is no special purpose when choosing the drain voltage of 0.1 V . Depending on the channel length (~4 μm in Fig. 2e), the CNTs in a single channel can withstand a voltage of no more than 10 V while a much higher drain voltage will possibly burn the CNTs in air.** There are two specific considerations for us to use this voltage in the scheme shown as Fig. 2e. Firstly, **the drain voltage of 0.1 V is a common safe parameter for testing CNTs as well as an optimal working condition with less switching power. At the same time, to operate at 0.1 V instead of other voltages can ensure safe operation by lowering the absolute current of the whole device.** Limited by the probe and apparatus of measurement, a drain voltage of higher than 0.1 V in our interdigitated devices may lead to current over 100 mA, which would approach the limit of our instrument and probes.

On the other hand, we haven't yet focused on the relationships between the bandgap of CNTs and the drain voltage. **It's pretty hard to testify such dependence through general characterization due to the difficulties in identifying the chiral indices of CNTs measured under transistors. Also, a narrower adjustment range of the drain voltage limits the research of mediating V_{DS} , as the instrument cannot withstand higher voltage and more powerful output.** But what you have proposed is so inspiring that the relationships if discovered may result in more significant insights into the field of CNT electronics. We appreciate your recommendations and directions in conducting further study in this field, especially about this important problem. **Actually, it's our intention as well to understand the complicated relationships and dependence behind, therefore, we have recently costed a remarkably high price to purchase a double-Cs corrected TEM just in order to analyze the microscopic CNT bandgap.** We hope you can leave us more time and look forward to our subsequent work in illustrating more fantastic phenomena of carbon-based electronics, especially on these ultrapure semiconducting ultralong CNTs.

We agree with you to depict the drain voltage and its associations with the CNT bandgap to attract more general readers. Impeded by our limited research, we're wondering if it's appropriate to revise the manuscript as follows (the red words are those after revising while the red words with the bold and underlining font are revisions aimed to this part).

...Depending on the measurement setup, it is possible to be sensitive to I_{DS} minus I_G when expecting to be sensitive only to I_{DS} , which can result in the artificially low measurement of the off current. However, the gate current is ~ 10 times lower than the off current so that the influence can be negligible on the pristine off current and the on/off ratio. The drain voltage of 0.1 V is a common safe parameter for testing CNTs as well as an optimal working condition with less switching power. Although few walled ultralong CNTs with smaller outermost bandgap were reported to carry the higher saturated current than that of SWNTs^{4,13}, the potential limit of the operating V_{DS} corresponding to the CNT bandgap requires further study.

Comment 6

The authors now state "It's worthy to note that there is no preference towards any kind of CNTs, single-walled (SWNTs), double-walled (DWNTs) or triple-walled (TWNTs) when discussing the rate selected growth." But what they mean is (I think) that their experiment is not sensitive to any preference. In fact, there is likely a significant difference, given the disappearance of the SWNTs.

Reply: We are grateful for your assistance in identifying and correcting this important problem. During the last revision process, we have suffered from the difficulty in how to express this issue accurately. We respect your expertise in understanding and analyzing the key problems. The following is the modification we're planning to make (the words in green were copied from the main text while the red words are those after revising).

In the main text

...Electron-donating surfactant treatment, which was verified efficient in strengthening BWF¹⁴, further confirmed the enrichment of s-CNTs longer than 154 nm (Supplementary Fig. 5). The aligned CNTs synthesized at other conditions (Supplementary Fig. 6) also demonstrated the ultrahigh s-purity at length >154 nm. It is significant to note that when discussing the rate selected growth, our measurements and statistics are not sensitive towards any kind of CNTs,

single-walled (SWNTs), double-walled (DWNTs) or triple-walled (TWNTs). While there might be a potential connection with the wall numbers that requires further study.

Comment 7

It would be clearer if at the point in the manuscript claiming 99.99% s-CNT purity (“A G mode collective (Supplementary Fig. 4) statistically counted from ~10⁴ tubes longer than 154 mm, displayed neither BWF nor D band signals”) mentioned explicitly that the careful Raman analysis is a summation of spectra from individual CNTs, where each spectrum is proven to be clearly sensitive to electron character, using the ED data. It’s clear in the SI, but not as worded in the main text.

Reply: We are deeply impressed by your strictness and specialization of wording. What you have proposed is obviously superior to our way of revising, which can best illustrate our experiments. By following your professional advice, it will definitely improve our work to avoid any misunderstanding to the potential readers (the words in green were copied from the main text while the red words are those after revising).

In the main text

...Then, almost all the m-CNTs will decay at a length of 154 mm, leaving the target 99.9999% s-CNTs (Fig. 1a). Careful Raman G mode analysis was conducted by collecting the spectra of ~10⁴ individual CNTs (Supplementary Fig. 4), where each spectrum was proven to be clearly sensitive to the CNT’s chiral structure (Supplementary Fig. 5). All the spectra displayed neither BWF nor D band signals, directly testifying the ultrapure s-CNTs (at least 99.99%).

Comment 8

The claim that the TOF is “also the highest among the reported industrial catalytic reactions.” needs some reference(s); is the effect just the higher temperature of the process, or something fundamental?

Reply: Thanks for your questions and remind. We feel so sorry for the lack of necessary references when mentioning the comparison of TOF with other reported industrial catalytic reactions. We have added the following references where the latest report of TOF for industrial catalysis is around ~10² s⁻¹, nearly four orders of magnitude lower than ours.

1. Ardagh, M. A., Abdelrahman, O. & Dauenhauer, P. J. Principles of Dynamic Heterogeneous Catalysis: Surface Resonance and Turnover Frequency Response. *ACS Catalysis* (2019).

2. Schmidt, L. D. *The Engineering of Chemical Reactions*. 536 (Oxford University Press: New York, 1998).

Actually, TOF, the turnover frequency is common in industrial catalysis. According to the descriptions in the textbook¹⁵, it quantifies the specific activity of a catalytic center for a special reaction under defined reaction conditions by the number of molecular reactions or catalytic cycles occurring at the center per unit time. The larger the TOF, the more active the catalyst. **Generally, for most relevant industrial applications the TOF is in the range of 10^{-2} – 10^2 s⁻¹, which is the common experience of chemical engineers and has been mentioned as well in that textbook.** In order to avoid causing any confusion, we'd like to make the following revision (the words in green were copied from the main text while the red words are those after revising).

In the main text

...S-CNTs define a much broader and higher space than m-CNTs versus the bandgaps. The average TOF of s-CNTs ($\sim 1.5 \times 10^6$ s⁻¹) is an order of magnitude higher than that of m-CNTs ($\sim 1.3 \times 10^5$ s⁻¹) and also the highest among the reported industrial catalytic reactions (10^{-2} – 10^2 s⁻¹ for most relevant industrial applications^{15,16}).

Comment 9

The Schrodinger's cat analogy does not make sense. The grown part of the tube exists unchanged whether or not the end is still growing. The tail of the cat is different depending on the quantum event, the tail of the CNT is the same. The analogy serves no purpose. The language is difficult to read throughout and would benefit from a thorough review.

Reply: We understand the point that you feel worried about. Actually, we consider the CNT growth as an interesting natural event analogous to the Schrodinger's cat. They both explicitly exhibit the coherence of the two definite states of life, alive and dead. The Schrodinger's cat is undoubtedly an entangled state while the carbon dimer addition onto the interface of each round is also an entangled state where the possibility of the parent CNT still alive is depicted as the catalyst activity probability. The difference is that more stable and persistent coherence

exists during the growth of an ultralong CNT even retaining along a long distance of 650 mm. Therefore, we figuratively call this as a huge Schrodinger's cat in order to attract the interests of more readers not only in the field of CNTs. It's also a prospect of our work in the near future and that's why we put it only as an ending of the text without more detailed descriptions. However, after all, we still lack enough evidence and words to illustrate this analogy. In order to avoid causing any misunderstanding, we'd like to remove it and depict it more deeply in the subsequent work.

In the main text

...Only ultralong s-CNTs with higher TOF and stable growth rates can survive in the final rate-competitive growth. This high-speed interlocking growth mode can even support 3.9×10^{10} steps of dimer-additions with equal survival probabilities and uniform time interval, ~~producing a seemingly huge 'Schrödinger-cat' state over a 650-mm-long distance.~~ We anticipate that similar behaviors are adaptable for other substances agreeing with the SF distribution.

[Summary]

We're grateful so much for your really patient and professional directions in improving our work, especially when you're trying to help seek the fundamental issues behind our technical experiments. We admire your insights and sense of responsibility as a reviewer for Nature Communications, which has set a respectful model for us. Through these two rounds of reviews, you can recognize our less sensitive sense in the physics of electronics despite a better understanding of the synthesis and characterization of materials. Your professional suggestions have definitely helped us make up for such disadvantages. We hope that all our efforts devoted to these two rounds of reviews can receive your approval and get this report qualified. After all, this might be the final chance for us. At the same time, we are expecting the precious chances of cooperating with experts like you to develop more advanced applications based on these highly pure semiconducting CNTs in the near future.

Reviewer #4

General comments

The paper has been improved. The updates have partially addressed the revisions that I previously requested. However, there are still deficiencies that remain to be addressed. Overall, my opinion is that this is a very interesting paper worth publishing in Nature Communications, provided these deficiencies can be addressed.

Reply: Thanks for your positive comments. It's our pleasure to obtain another chance to improve our work further according to your professional suggestions. We're sincerely expecting to get this report qualified and satisfied after this round of revision as this might be the final chance for us.

Follow-up on previous comment 1)

The misleading reporting of current per width for the interdigitated arrays has been eliminated in the abstract. However, it still appears in the y-axis of Fig. 2h, in Fig. 2f as the solid red stars, and in the text. Please update as follows:

a) Update the data shown in Fig. 2h y-axis to report current per width where the width is the physical width of each channel times the number of interdigitated channels. b) Remove the solid red stars in Fig. 2f.

Reply: We appreciate so much for your patient and detailed directions in improving our manuscript. According to your suggestions, **we have modified the data shown in Fig. 2h, just reporting the single-channel device performances so as to avoid any misunderstanding. Also, the data symbolled by solid red stars have been removed and we labeled that area as an outstanding 'superb', which indicated an expected target we hope to achieve in the near future.**

c) Fix the sentence "At a drain-source bias (VDS) of 0.1 V, the device delivered a high current of 1.4 mA/um", which does not make sense because the channel width is actually much more. The authors follow up this sentence with "(corresponding to ~14 uA/um for single-channel..." It is this 14 uA/um number that should be reported in the first place.

Reply: Thanks for your kind remind. We admire what you have addressed and it would be more accurate if we describe the results like this. Besides, **we have reported the absolute current**

for the whole device instead of the width-normalized one for 100 channels. The revision has been shown as follows (the words in green were copied from the main text while the red words are those after revising).

...Even though the CNTs were densified by multiple growths, transistors fabricated on those tubes longer than 154 nm all demonstrated high on/off ratio after each cycle (Supplementary Fig. 9). Prototypical ~100 parallel transistors (Figs. 2d-g) were fabricated on aligned long tubes contacted with interdigitated electrodes. At a drain-source bias (V_{DS}) of -0.1 V, the device delivered a high width-normalized current of ~14 $\mu\text{A}/\mu\text{m}$ for a single channel (corresponding to ~42 mA output for the whole device) with an on/off ratio of 10^8 and field effect mobility¹² >4000 $\text{cm}^2/\text{V}\cdot\text{s}$ (Fig. 2h).

The authors' argument that '...Compared with single-channel devices, this long-range interdigitated scheme fully utilizes the length strength of ultralong CNTs and compensates for the lower current output caused by lower density' is misleading. The increase in current with more interdigitations is NOT a property unique to their ultralong CNTs. The current of any material: silicon, oxides, graphene, 2D materials, random networks of CNTs, aligned arrays of short CNTs - all will increase using the interdigitation scheme linearly with the number of electrodes. There is no limit to this effect - the more interdigitations the more the current. This "strategy" of using interdigitations to artificially increase current per width has no merit and would only harm the field and scientific if established as a means for making device performance *seem* better.

Reply: Thanks to your professional comment, we have realized that it's definitely not a property unique to ultralong CNTs available to high output delivery with the interdigitated scheme. Our statement, '...Compared with single-channel devices, this long-range interdigitated scheme fully utilizes the length strength of ultralong CNTs and compensates for the lower current output caused by lower density' will possibly mislead our general readers. We had intended to highlight the length strength of these ultralong CNTs in devices, but it seems to bring more dispute than significance with this controversial claim. In order to make it more easily understood, **we'd like to follow your advice to remove such statements** . Besides, **we'd like to provide additional interpretations about the significance of interdigitated scheme**. It's not our purpose to emphasize the high output delivery in this work, because we admit that the

density of ultralong CNTs as synthesized is not high enough. **But with this scheme covering millimeter-long CNTs, the device can still retain a high on/off ratio and mobility. We believe that it's another strong evidence demonstrating the stable semiconducting property of these long tubes.** Therefore, we'd like to supplement additional statements as follows in order to give more insights on the improved semiconducting purity, perfect structures and electrical performances (the red words are those after revising while the red words with the bold and underlining font are revisions aimed to this part).

...At a drain-source bias (V_{DS}) of -0.1 V, the device delivered a high width-normalized current of $\sim 14 \mu\text{A}/\mu\text{m}$ for a single channel (corresponding to $\sim 42 \text{ mA}$ output for the whole device) with an on/off ratio of 10^8 and field effect mobility¹² $> 4000 \text{ cm}^2/\text{V}\cdot\text{s}$ (**Fig. 2h**)....

Covering 1.2-mm-long CNTs, this interdigitated device can still retain a high on/off ratio and mobility, which effectively demonstrates the stable semiconducting property and perfect structure of the long CNTs.

The ability to determine a higher on/off ratio is still true if both on and off current are properly normalized by the actual channel width (width times the number of channels in parallel)... so this is not justification for using the falsely normalized current per width.

Reply: We understand your point and thanks for your kind remind. **Actually, both on and off current in our last report have been normalized by the actual channel width covering conductive CNTs.** But they have not been normalized by the number of channels. In this round of review, **we have taken your advice to modify our data in Fig. 2h, only reporting the results for a single channel so that it can be fairly compared to other devices previously reported.**

Follow up on previous comment 5)

Without a good justification of how the on/off ratio is so high, the authors should at minimum provide more information about their measurement setup and its abilities to ensure the reader and community that their measurements of off-current are not being skewed by unexpected factors. Please (a) provide measures of the gate current I_g as a function of V_{gs} in devices in which I_{ds} is also measured as a function of V_{gs} . Depending on measurement setup, it is possible to be sensitive to I_{ds} minus I_g when expecting to be sensitive only to I_{ds} , which can result in artificially low measures of I_{off} . Miscalibration (Which can be tested in FET devices without

nanotubes) should also be characterized.

Reply: Thanks for your kind remind. Actually, we also feel surprised why such high on/off ratio can be achieved on the transistors fabricated on these few-walled CNTs with smaller bandgap. Your professional suggestions should help eliminate the controversy and confusion from our readers. We agree with you to provide more information about the measurement setup. Firstly, **we extracted the I_G - V_G data corresponding to the device as shown in Fig. 2h and found that the I_G fluctuated around $10 \text{ pA}/\mu\text{m}$, although we had expected the I_G to be around zero.** On the one hand, we don't believe that it originates from the zero error of the setup according to the precise miscalibration record in Fig. S10c. The zero current characterized in the FET device without CNTs indicates the less error of this setup despite some predictable ambient noise. At the same time, it couldn't be the result of gate current leakage. To make it clearer, we have especially discussed with some semiconductor scholars at Tsinghua University. They have reached a consensus that **the gate with leakage would behave as a resistor conforming to the Ohm's Law with possible barrier characteristic, instead of retaining the current direction during the voltage sweeping.** Therefore, we reasonably believe that there is no obvious current leakage from the gate. **As for the biased current, we suppose that it might be caused by the capacitance, of which the absolute value depends on multiple factors like scanning rate, scanning direction, etc. The lower I_G near the current valley can be attributed to this capacitance effect as well, given the lower scan rate in response to the ultralow current.** Besides, in order to rule out the contingency, **we compared this result with some devices fabricated on single ultralong CNTs and a similar biased current was found as well (Fig. S10b).** However, for any reason, **such bias is ~ 10 times lower than the off current, which will cause negligible influence on the pristine off current. The actual on/off ratio can still be higher than 10^8 .** Additionally, the supplementary I_G - V_G data and miscalibration record can ensure our readers that there is no obvious error in our setup. It requires a detailed study to explore the specific reasons for this biased current and how capacitance affects the electrical characteristics. We hope that our subsequent work can be expected to discuss these problems more deeply. For this work, we'd like to supplement the following information (the words in green were copied from the main text while the red words are those after revising, the red words with the bold and underlining font are revisions aimed to this part).

In Methods

Fabrication and measurement of the transistors. Ultralong CNTs were grown on a heavily n-doped silicon wafer with an 800-nm SiO₂ top layer..... Electronic measurements were carried out by applying drain and gate voltages relative to the source electrode with a Keithley 4200A parameter analyzer at room temperature in air. The drain current was measured with a Keithley 4200-PA amplifier. The low current measurement capabilities of any SMU can be extended by adding an optional 4200-PA preamplifier. The preamplifier provides 10 aA resolution by effectively adding five current ranges to either SMU model.

In the main text

...At a drain-source bias (V_{DS}) of -0.1 V, the device delivered a high width-normalized current of $\sim 14 \mu\text{A}/\mu\text{m}$ for a single channel (corresponding to $\sim 42 \text{ mA}$ output for the whole device) with an on/off ratio of 10^8 and field effect mobility¹² $>4000 \text{ cm}^2/\text{V}\cdot\text{s}$ (Fig. 2h). There is no obvious leakage current from the gate while the device is being operated. And the transfer characteristic has been characterized when there are no CNTs working in the channel (Supplementary Fig. 10). Depending on the measurement setup, it is possible to be sensitive to I_{DS} minus I_G when we expect it to be sensitive only to I_{DS} , which can result in the artificially low measurement of the off current. However, the gate current is ~ 10 times lower than the off current so that the influence of gate current can be negligible on the pristine off current and the actual on/off ratio.

In Supporting information

Supplementary Fig. 10 | **a**, the gate current I_G versus V_{GS} of the device shown in Fig. 2h where I_{DS} is also measured as a function of V_{GS} . The current is width-normalized with applied V_{DS} of -0.1 V. I_G couldn't be the result of leakage from the gate. Because it would behave as a resistor conforming to the Ohm's Law when there is current leakage, instead of retaining the positive current direction during the voltage sweeping. We suppose that it might be caused by the capacitance, of which the absolute value depends on multiple factors like scanning rate, scanning direction, etc. The lower I_G near the current valley can be attributed to this capacitance effect as well, given the lower scan rate in response to the ultralow current. **b**, Transfer characteristic of a transistor fabricated on one ultralong CNT plotted in logarithmic scale with applied V_{DS} of -0.5 V. The corresponding I_G versus V_{GS} data is also shown in yellow. **c**, Transfer characteristic measured in transistors without CNTs.

New comment 9)

Some of the new sentences need more editing to improve readability and grammar including.

“Besides, the resonance of these long tubes with the lasers was confirmed again from the correspondence of m-CNT (s-CNT) quantities between BWF (Lorentzian) shaped peaks and chirality identification under TEM (Supplementary Figs. 12-13).”

“Despite that the lasers we used can excite the ultralong CNTs within limited diameter distributions, the laser conditions, CNT structures, overlapping peaks, etc. will all influence the final observed numbers of components¹⁹.”

Reply: Thanks for pointing out these linguistic problems. In the process of revising the manuscript, we neglected the importance of readability and grammar, resulting in some unauthentic Chinglish. We hope that the following corrections and adjustment can be clearer and more readable (the words in green were copied from the main text while the red words are those after revising).

...The average TOF of s-CNTs ($\sim 1.5 \times 10^6 \text{ s}^{-1}$) is an order of magnitude higher than that of m-CNTs ($\sim 1.3 \times 10^5 \text{ s}^{-1}$) and also the highest among the reported industrial catalytic reactions. Besides, the fact that the numbers of m-CNTs (or s-CNTs) are identical between the results of Raman spectra and electron diffraction, proves the assumption that these long tubes just resonated with the laser irradiation wavelength we used (Supplementary Fig. 15).

...Although within a narrow range of diameter, the ultralong CNTs could resonate with the laser irradiation wavelength we used, the actual number of the G mode components that can be observed depends on the laser power, chiral structures of CNTs, etc.¹⁷.

New comment 10)

Possible typos in:

Fig. S10d caption “d, The histogram showing the distribution of on-state conductance as measured in c. e, Collection of output characteristics with an applied VGS of 15 V”. 15 V should be -15 V, presumably.

“b, Output characteristic of the transistor measured with ascending VGS from 11 V to 17 V at a step of 1 V.” should be -11 to -17 at a step of -1 V, presumably.

Reply: It's our honor and pleasure to receive your times of patient assistance in correcting our mistakes. We feel so sorry to bring you much trouble due to our carelessness and we have modified them by adjusting their units as follows. Please give us some understanding of the lack of expertise in electronics while focusing more on material synthesis. We have modified the manuscript as follows (the words in green were copied from the main text while the red words are those after revising).

In Supporting Materials

Supplementary Fig. 11 | Electrical performances of a representative transistor. b, Output characteristic of the transistor measured with ascending V_{GS} from -11 V to -17 V at a step of -1 V.

Supplementary Fig. 12 | Statistics of the electrical performances of single-tube transistors. e, Collection of output characteristics with an applied V_{GS} of -15 V.

[Summary]

Certainly, we have benefited much from all you have addressed in these two rounds of reviews, which will enlighten our work in the future and we'd like to conduct deep research, and if possible, to cooperate with experts in electronics like you, in the hope of developing more advanced devices with these perfect ultralong CNTs. This might be our final chance to improve our work by following your advice. Thanks again for your patience and professional directions. Best wishes to you and wish you all the best!

References

- 1 Hu, Y. *et al.* Growth of high-density horizontally aligned SWNT arrays using Trojan catalysts. *Nat Commun* **6** (2015).
- 2 Zhang, F. *et al.* Growth of semiconducting single-wall carbon nanotubes with a narrow band-gap distribution. *Nat Commun* **7** (2016).
- 3 Geblinger, N., Ismach, A. & Joselevich, E. Self-organized nanotube serpentines. *Nat Nanotechnol* **3**, 195-200 (2008).
- 4 Zhu, Z. *et al.* Acoustic-assisted assembly of an individual monochromatic ultralong carbon nanotube for high on-current transistors. *Sci Adv* **2** (2016).
- 5 Javey, A., Guo, J., Wang, Q., Lundstrom, M. & Dai, H. Ballistic carbon nanotube field-effect transistors. *Nature* **424**, 654 (2003).
- 6 Brady, G. J. *et al.* Quasi-ballistic carbon nanotube array transistors with current density exceeding Si and GaAs. *Sci Adv* **2** (2016).
- 7 Cao, Q. *et al.* Arrays of single-walled carbon nanotubes with full surface coverage for high-performance electronics. *Nat Nanotechnol* **8**, 180-186 (2013).
- 8 Akinwande, D., Nishi, Y. & Wong, H. P. An Analytical Derivation of the Density of States, Effective Mass, and Carrier Density for Achiral Carbon Nanotubes. *IEEE T Electron Dev* **55**, 289-297 (2008).
- 9 Artyukhov, V. I., Penev, E. S. & Yakobson, B. I. Why nanotubes grow chiral. *Nat Commun* **5**, 4892 (2014).
- 10 Elliott, J. A., Yasushi, S., Hakim, A., Christophe, B. & Neyts, E. C. Atomistic modelling of CVD synthesis of carbon nanotubes and graphene. *Nanoscale* **5**, 6662-6676 (2013).
- 11 Zhang, R., Zhang, Y. & Wei, F. Controlled Synthesis of Ultralong Carbon Nanotubes with Perfect Structures and Extraordinary Properties. *Acc Chem Res* **50**, 179-189 (2017).
- 12 Zhang, Z. *et al.* Almost Perfectly Symmetric SWCNT-Based CMOS Devices and Scaling. *ACS Nano* **3**, 3781-3787 (2009).
- 13 Wen, Q. *et al.* 100 mm Long, Semiconducting Triple-Walled Carbon Nanotubes. *Adv Mat* **22**, 1867-1871 (2010).
- 14 Blackburn, J. L., Engtrakul, C., McDonald, T. J., Dillon, A. C. & Heben, M. J. Effects of Surfactant and Boron Doping on the BWF Feature in the Raman Spectrum of Single-Wall Carbon Nanotube Aqueous Dispersions. *J Phy Chem B* **110**, 25551-25558 (2006).
- 15 Schmidt, L. D. *The Engineering of Chemical Reactions*. 536 (Oxford University Press: New York, 1998).
- 16 Ardagh, M. A., Abdelrahman, O. & Dauenhauer, P. J. Principles of Dynamic Heterogeneous Catalysis: Surface Resonance and Turnover Frequency Response. *ACS Catalysis* (2019).
- 17 Levshov, D. I. *et al.* Accurate determination of the chiral indices of individual carbon nanotubes by combining electron diffraction and Resonant Raman spectroscopy. *Carbon* **114**, 141-159 (2017).

Reviewers' Comments:

Reviewer #4:

Remarks to the Author:

Most of the sticking points have now been addressed. Frankly, the new text that has been added has not been written as clearly as the rest of the pre-existing text in the manuscript. Nonetheless, the paper has been improved in the following ways:

- 1) The paper has now finally removed the current density values that were misleading due to the interdigitation effect.
- 2) Reviewer #2 has made an important request that the paper state that the 99.9999% semiconducting nanotube purity is an extrapolated or predicted quantity. The revised paper has now added the wording "the predicted" to the abstract to satisfy this request. Although the exact wording of the sentence in the abstract is difficult to understand, the idea that this is an extrapolated value is clear after reading the paper in its entirety.
- 3) The paper now acknowledges upfront that the large on/off ratio that is measured is somewhat unexpected given the small bandgap of the outer most nanotubes. It seems that neither the reviewers or the authors have a good explanation for why the on/off ratio is so high. However, as long as the reviewers acknowledge upfront this unexpected result then I am fine with it. To me, the main result of this paper revolves around synthesis not the electrical data – and the characterization of the synthesis is more complete and compelling.

The paper can be publishable after addressing the following minor comments:

- 4) The paper has added the following sentence, "We assume that the energy relationship is not enough for Dirac carbon species¹³, and additional dimensionality of momentum might be another important factor we had not expected." However, I have no idea what this sentence means! It just does not make sense to me. Please revise to clarify or remove.
- 5) The abstract uses the phrasing "entangled condensate" which has a very specific meaning in quantum fields that is not applicable here. Please use different wording.
- 6) "The paper now states that the gate current is ~10 times lower than the off current so that the influence of gate current can be negligible on the pristine off current and the actual on/off ratio." However, Fig. S10a,b clearly shows this statement is not true, in which I_G is similar to or greater than I_D . Please revise this statement.
- 7) If I_G is non-zero (due to resistive transport through the oxide or due to $C \cdot dV/dt$) then there must be an equal but opposite current flowing out of I_D or I_S or a combination of both, which depending on the measurement setup, can affect the quantification of the off-state current and the on/off ratio. Please revise estimate of on/off ratio if needed.

Response to Reviewers' Comments

Reviewer #4

General comments

Most of the sticking points have now been addressed. Frankly, the new text that has been added has not been written as clearly as the rest of the pre-existing text in the manuscript. Nonetheless, the paper has been improved in the following ways:

Reply: Thanks for your warmhearted evaluation and feedback. We'd like to improve our manuscript according to your professional suggestions.

Comments 1-3

1) The paper has now finally removed the current density values that were misleading due to the interdigitation effect.

2) Reviewer #2 has made an important request that the paper state that the 99.9999% semiconducting nanotube purity is an extrapolated or predicted quantity. The revised paper has now added the wording "the predicted" to the abstract to satisfy this request. Although the exact wording of the sentence in the abstract is difficult to understand, the idea that this is an extrapolated value is clear after reading the paper in its entirety.

3) The paper now acknowledges upfront that the large on/off ratio that is measured is somewhat unexpected given the small bandgap of the outer most nanotubes. It seems that neither the reviewers or the authors have a good explanation for why the on/off ratio is so high. However, as long as the reviewers acknowledge upfront this unexpected result then I am fine with it. To me, the main result of this paper revolves around synthesis not the electrical data – and the characterization of the synthesis is more complete and compelling.

Reply: Thanks for your recognition and evaluation about our recent revision. It's our pleasure to learn from professional reviewers like you, about these skills on how to make our manuscript more rigorous. We're delighted to see that our efforts on this manuscript can be finally recognized under these reviewers' patient directions.

Comment 4

The paper has added the following sentence, "We assume that the energy relationship is not enough for Dirac carbon species¹³, and additional dimensionality of momentum might be another important factor we had not expected." However, I have no idea what this sentence means! It just does not make sense to me. Please revise to clarify or remove.

Reply: Thanks for your advice. Actually, we had intended to propose one of our assumptions, as there might be a physical theory bridging the gap between atomistic dynamics and macroscopic scales. But as you mentioned, it still requires sufficient evidence to demonstrate this assumption. In order to avoid any misunderstanding, we've decided to remove this statement.

Comment 5

The abstract uses the phrasing "entangled condensate" which has a very specific meaning in quantum fields that is not applicable here. Please use different wording.

Reply: Thanks for your kind remind. We were just making a comparison with the entangled state in the quantum field. But as you said, it would result in unnecessary misunderstandings. We have changed it as 'perfectly assembled nanotubes' with a clearer meaning.

Comments 6-7

6) "The paper now states that the gate current is ~10 times lower than the off current so that the influence of gate current can be negligible on the pristine off current and the actual on/off ratio." However, Fig. S10a,b clearly shows this statement is not true, in which I_G is similar to or greater than I_D . Please revise this statement.

7) If I_G is non-zero (due to resistive transport through the oxide or due to $C \cdot dV/dt$) then there must be an equal but opposite current flowing out of I_D or I_S or a combination of both, which depending on the measurement setup, can affect the quantification of the off-state current and the on/off ratio. Please revise estimate of on/off ratio if needed.

Reply: Thanks for pointing out this problem. Actually, what we'd like to emphasize is the stable off current, the tail of the transfer characteristic, which is almost 10 times higher than the gate current. But it seems not obvious enough to directly recognize due to their similar values. Just as you said, we agree that the non-zero gate current is caused by capacitance, but it won't have an outstanding impact on the on/off ratio. The areal off current can be estimated as 'measured off current \pm measured gate current'. But the impact on the areal on/off ratio will be negligible,

as the errors should be around (0.5~1) times of on/off ratio, which still enables a high on/off ratio. In order to make it clearer, we'd like to make the following revision (the words in green were copied from the main text while the red words are those after revising).

Main text

.....There is no obvious leakage current from the gate while the device is being operated. The impact of the gate current can be neglected on the areal on/off ratio. And the transfer characteristic has been characterized when there are no CNTs working in the channel (Supplementary Figure 10). ~~Depending on the measurement setup, it is possible to be sensitive to I_{DS} minus I_G when we expect it to be sensitive only to I_{DS} , which can result in the artificially low measurement of the off current. However, the gate current is ~10 times lower than the off current so that the influence of gate current can be negligible on the pristine off current and the actual on/off ratio.~~ The drain voltage of 0.1 V is a common safe parameter for testing CNTs as well as an optimal working condition with less switching power.

Supporting Information

Supplementary Figure 10 | a, The gate current I_G versus V_{GS} of the device shown in Fig. 2h where I_{DS} is also measured as a function of V_{GS} . The current is width-normalized with applied V_{DS} of -0.1 V. I_G couldn't be the result of leakage from the gate. Because it would behave as a resistor conforming to the Ohm's Law when there is current leakage, instead of retaining the positive current direction during the voltage sweeping. We suppose it might be caused by the capacitance, of which the absolute value depends on multiple factors like scanning rate, scanning direction, etc. The lower I_G near the current valley can be attributed to this capacitance effect as well, given the lower scan rate (longer integration time) in measuring the ultralow current. I_G have a limited impact on the on/off ratio due to its similar values to the stable off current. The areal off current can be estimated as 'measured off current \pm gate current', which still enables the on/off ratio higher than 10^8 .

[Summary]

Thanks for providing your professional insights on our manuscript. It has definitely given us much inspiration not only on improving this work but also enlightening our directions in the future. We'll devote more efforts in developing advanced technology based on these perfect ultralong carbon nanotubes.